# The Nuclear Route: Sharp Asymptotics of ERM in Overparameterized Quadratic Networks

**Vittorio Erba**
Statistical Physics of Computation Laboratory
EPFL, Switzerland

**Emanuele Troiani**
Statistical Physics of Computation Laboratory
EPFL, Switzerland

**Lenka Zdeborová**
Statistical Physics of Computation Laboratory
EPFL, Switzerland

**Florent Krzakala**
Information, Learning & Physics Laboratory
EPFL, Switzerland

## Abstract

We study the high-dimensional asymptotics of empirical risk minimization (ERM) in over-parametrized two-layer neural networks with quadratic activations trained on synthetic data. We derive sharp asymptotics for both training and test errors by mapping the $\ell_2$-regularized learning problem to a convex matrix sensing task with nuclear norm penalization. This reveals that capacity control in such networks emerges from a low-rank structure in the learned feature maps. Our results characterize the global minima of the loss and yield precise generalization thresholds, showing how the width of the target function governs learnability. This analysis bridges and extends ideas from spin-glass methods, matrix factorization, and convex optimization and emphasizes the deep link between low-rank matrix sensing and learning in quadratic neural networks.

Modern machine learning relies heavily on training highly over-parameterized neural networks, which often generalize well despite having far more parameters than data points [Zhang et al., 2021]. While it is known that large non-linear networks can approximate many functions [Cybenko, 1989], it remains unclear what these models actually learn in practice, and why training succeeds so often. In particular, we lack a precise understanding of how the structure of the data and the target function affects learnability, and how many samples are needed. Developing such an understanding remains a central theoretical challenge.

A promising route toward addressing these questions is to study models that go beyond the linear case—already well understood, eg. [Lee et al., 2018, Ji and Telgarsky, 2019, Pesme et al., 2021]—while remaining simple enough for rigorous analysis. Two-layer networks with quadratic activations are then the next natural candidate that captures some nonlinear behavior while still allowing for mathematical treatment [Gamarnik et al., 2019, Arjevani et al., 2025, Maillard et al., 2024]. On top of that working with synthetic data helps isolate the core mechanisms of learning and generalization [Mei et al., 2018, Aubin et al., 2020, Mei and Montanari, 2022, Xiao et al., 2022, Damian et al., 2024], free from the confounding factors of real-world datasets.

In this paper, we thus study learning by empirical risk minimization with quadratic networks from data that is also generated by a quadratic network. Consider a dataset $\mathcal{D} = \{\mathbf{x}^\mu, y^\mu\}_{\mu=1}^n$ where the data $\mathbf{x}^\mu \in \mathbb{R}^d$ are standard Gaussian $\mathbf{x}^\mu \sim \mathcal{N}(0, \mathbb{I}_d)$ (though our results allow for some universality) for $\mu = 1, \ldots, n$, and the labels $y^\mu \in \mathbb{R}$ are generated by an unknown *target* function $f^\star(\mathbf{x})$. We aim at learning this unknown function using a quadratic neural network $\hat{f}(x; W)$ with a number of

hidden unit $m \geq d$:

$$\hat{y} = \hat{f}(\mathbf{x}; \hat{W}) := \frac{1}{\sqrt{m}} \sum_{k=1}^{m} \sigma_k \left( \frac{\hat{\mathbf{w}}_k \cdot \mathbf{x}}{\sqrt{d}} \right) , \quad (1)$$

where $\sigma_k(u) = u^2 - ||\mathbf{w}_k||^2/d$ is the (centered) quadratic activation, and where we collect the first-layer weights $\mathbf{w}_k \in \mathbb{R}^d$ for $k = 1, \ldots, m$ in the matrix $W \in \mathbb{R}^{m \times d}$. We learn $W$ by empirical risk minimization of the square loss with $\ell_2$ regularization (or equivalently, weight decay):

$$\hat{W} = \arg\min \mathcal{L}(W), \quad \text{where} \quad \mathcal{L}(W) := \sum_{\mu=1}^{n} \left( y^\mu - \hat{f}(\mathbf{x}^\mu; W) \right)^2 + \lambda \|W\|_F^2 . \quad (2)$$

Given the structure of the model, the considered quadratic neural network can represent any centered positive semi-definite quadratic form of the input data—but no more. In particular, functions involving higher-order nonlinearities cannot be captured and are effectively treated as noise by the learner. To focus on the regime where generalization is possible, we therefore choose a target function $f^\star$ that lies within the expressivity class of the model (1) (we will also refer to the model as the *student* while thinking about the target as the *teacher*):

$$y^\mu = f^*(\mathbf{x}^\mu; W^*) + \sqrt{\Delta}\xi^\mu , \quad \text{with} \quad f^*(\mathbf{x}; W^*) := \frac{1}{\sqrt{m^*}} \sum_{k=1}^{m^*} \sigma_k \left( \frac{\mathbf{w}_k^\star \cdot \mathbf{x}}{\sqrt{d}} \right) , \quad (3)$$

where $\xi^\mu \sim \mathcal{N}(0,1)$ is an additional Gaussian label noise. As will shall see, our result will depend on $W^\star$ only through the spectral density of $S^\star = (W^\star)^T W^\star / \sqrt{m^\star d} \in \mathbb{R}^{d \times d}$, which needs to have a well-defined limit as $d \to \infty$.

We will work in the high-dimensional limit $d \to \infty$ with extensive-width target and quadratically many samples, i.e. the joint limit $d, n, m^*, m \to +\infty$ with

$$\alpha = n/d^2 = \mathcal{O}(1), \quad \kappa^* = m^*/d = \mathcal{O}(1), \quad \kappa = m/d = \mathcal{O}(1). \quad (4)$$

**Our contributions.** We provide an exact characterization of training and generalization in over-parameterized two-layer neural networks with quadratic activations, in the high-dimensional limit with Gaussian data. Our main result (Theorem 1) gives closed-form expressions for the training loss, generalization error, and spectral properties of the global minima of the regularized empirical risk (2), in the regime $\kappa = m/d \geq 1$, $\kappa^* = \mathcal{O}(1)$, and $\lambda > 0$.

Our solution to this problem connects three previously distinct lines of research that turn out to be related: the geometry and training dynamics of quadratic networks in teacher-student setups [Sarao Mannelli et al., 2020, Gamarnik et al., 2019, Martin et al., 2024a, Arjevani et al., 2025], recent advances in high-dimensional Bayesian analysis of networks with extensive width [Maillard and Bandeira, 2023, Maillard et al., 2024], and the role of implicit regularization in matrix factorization [Gunasekar et al., 2017] and its connection to matrix compressed sensing and nuclear norm regularization [Recht et al., 2010, Fazel et al., 2008].

Concretely, we map the non-linear estimation problem for $W$ in (2) to the linear one of estimating the matrix $S = W^\top W / \sqrt{md} \in \mathbb{R}^{d \times d}$, where, remarkably, the $\ell_2$ regularization on $W$ translates into a nuclear norm regularization on $S$. This reveals that the learning dynamics implicitly favor solutions $\hat{f}$ corresponding to narrow neural networks. We then study this equivalent matrix model by rigorous tools based on approximate message passing [Berthier et al., 2020, Gerbelot and Berthier, 2023] and their relation to convex optimization [Loureiro et al., 2021b]. Our main result is Theorem 1, an analytical prediction for the test error achieved by the ERM, where importantly the error does not depend on the value of $\kappa$ as long as $\kappa > 1$, thus describing potentially massively over-parametrized models. Theorem 1 allows us to answer a range of questions as a function of $\kappa^*$ (see Section 3).

1. Location of the interpolation threshold: How many samples $n$ are needed to have only a unique global minimum of (2) in the limit $\lambda = 0^+$, both in the noiseless and noisy case?

2. Generalization performance: How many samples $n$ are needed to achieve perfect generalization with zero label noise?

3. Low-rank limit: What is the generalization performance in the limit $\kappa^* \ll 1$ corresponding to low-rank target functions?

In this paper, we provide new sharp results focusing on the extensive width case, encompassing the full range of target widths and focusing in particular on the setting $0 < \kappa^* < 1$, previously unexplored.

Finally, we remark that while (2) is *a priori* non-convex, it has no non-global local minima [Arjevani et al., 2025], implying that if a gradient-based algorithm converges to a minimum at all, it must converge to a global one. Hence, our main result Theorem 1 provides *a closed-form characterization of the behavior at convergence to minima of gradient-based algorithms* for any strictly positive regularization $\lambda$.

**Further related work.** Deriving **exact formulas for two-layer networks** in the teacher-student setting in high dimensions is a classical problem that has been extensively studied over the past decades (e.g., [Gardner and Derrida, 1989, Bayati et al., 2010, Thrampoulidis et al., 2018, Barbier et al., 2019, Montanari et al., 2023]). The case of a single hidden unit, or single-index model, has been widely explored and, for quadratic activations, reduces to phase retrieval [Mondelli and Montanari, 2018, Lu and Li, 2020, Maillard et al., 2020]. The case of a finite number of hidden units falls into the broader class of multi-index models, which has seen a surge of recent interest [Seung et al., 1992, Aubin et al., 2018, Damian et al., 2022, Bietti et al., 2025, Troiani et al., 2025, Bruna and Hsu, 2025, Montanari and Urbani, 2025]. When the number of directions becomes extensive (i.e., of order $\mathcal{O}(d)$) and for general activations, the problem has been studied for linearly-many samples $n = \mathcal{O}(d)$ [Li and Sompolinsky, 2021, Cui et al., 2025, Pacelli et al., 2023, Bordelon and Pehlevan, 2023, Camilli et al., 2025]. It becomes significantly more challenging in the setting we consider here of quadratically-many samples $n = \mathcal{O}(d^2)$. This more difficult regime has been explored empirically [Cui et al., 2025] and heuristically [Barbier et al., 2025] in a few specific settings, but an asymptotically exact sharp characterization is still lacking.

Significant progress, however, has recently been made for the quadratic networks in the **Bayes-optimal, information-theoretic setting**. In particular, [Maillard et al., 2024] computed the Bayes-optimal performance in the setting considered here, a result later rigorously proven in [Xu et al., 2025]. These works build upon recent advances originating in the context of ellipsoid fitting [Maillard and Kunisky, 2024, Maillard and Bandeira, 2023], and leverage a Gaussian universality principle that also underpins our approach [Goldt et al., 2022, Hu and Lu, 2022, Montanari and Saeed, 2022, Dubova et al., 2023]. This allows to study the problem as a matrix denoising task [Bun et al., 2014, Troiani et al., 2022, Pourkamali and Macris, 2023a,b, Semerjian, 2024]. For **empirical minimization** instead, landscape properties have been studied in [Du and Lee, 2018, Venturi et al., 2019, Soltanolkotabi et al., 2019, Gamarnik et al., 2019, Sarao Mannelli et al., 2020, Arjevani et al., 2025], and gradient descent on the population loss has been discussed in [Martin et al., 2024b].

The concept of **double descent** (see e.g. [Belkin et al., 2019, Nakkiran et al., 2021]) has reshaped our understanding of the bias-variance trade-off, revealing that increasing model complexity can, counterintuitively, improve generalization. While many of these analyses have focused on linear or kernel regimes [Gerace et al., 2020, Geiger et al., 2020, Belkin et al., 2020, Mei and Montanari, 2022], our work instead provides sharp asymptotics for non-linear two-layer nets.

Our analysis for two-layer networks hinges on a mapping to a **matrix compressed sensing, or low rank-matrix recovery**, problem with nuclear norm regularization [Fazel et al., 2008]. The number of samples required for perfect recovery has been discussed in [Donoho et al., 2013, Oymak and Hassibi, 2016, Amelunxen et al., 2014]. Additionally, we leverage tools from **Approximate Message Passing (AMP)** theory. AMP has become a tool of choice for the rigorous analysis of statistical inference problems and algorithms in high-dimension. It often allows to provides sharp asymptotic characterizations of both Bayesian and empirical risk minimization procedures in many models [Bayati and Montanari, 2011, Donoho and Montanari, 2016, Rangan et al., 2016, Loureiro et al., 2021b, Vilucchio et al., 2025]. In this work, we use AMP with non-separable denoisers [Berthier et al., 2020, Gerbelot and Berthier, 2023] to prove rigorously our main formula, that can also be derived by the heuristic replica method, a non-rigorous technique from statistical physics [Mézard et al., 1987, Mezard and Montanari, 2009].

# 1 Setting and notations

We consider a dataset of $n$ samples $\mathcal{D} = \{\mathbf{x}^\mu, y^\mu\}_{\mu=1}^n$, with $\mathbf{x}^\mu \in \mathbb{R}^d$ and $y^\mu \in \mathbb{R}$, constructed as in (1). We remark that the assumption of Gaussian data could be relaxed in the same spirit as in [Xu

et al., 2025, Assumption 2.2], and denote by $\mathbb{E}$ the average of the training set. We will assume that the empirical spectral density of the matrix $S^* = (W^*)^T W^* / \sqrt{m^* d} \in \mathbb{R}^{d \times d}$ converges to a limiting distribution $\mu^*$ with finite first and second moment as $d \to \infty$ with $\kappa^* = m^*/d = \mathcal{O}(1)$, and call $Q^*$ its second moment. As our running example, we will focus on the *Marchenko-Pastur (MP) target* case, where the weights $\mathbf{w}^*$ are such that the limiting distribution satisfies $\mu^*(x) = \sqrt{\kappa^*} \mu_{\mathrm{M.P.}}(\sqrt{\kappa^*} x)$, where $\mu_{\mathrm{M.P.}}$ is the Marchenko-Pastur distribution [Marchenko and Pastur, 1967] with parameter $\kappa^*$ (the asymptotic spectral distribution of $A^T A / m$ where $A \in \mathbb{R}^{m \times d}$ has i.i.d. zero-mean unit-variance components), and $Q^* = 1 + \kappa^*$. For example, this is the case for $\mathbf{w}^*$ with i.i.d. components extracted from a distribution with zero mean and unit variance, but we stress that our results hold also for deterministic targets, as well as targets with different spectral distributions.

We learn the dataset by empirical risk minimization on the loss (2), and unless stated otherwise, in this paper we will always consider learning with $m \geq d$, typically for $m > m^*$. We will measure the performance of the empirical risk estimator using the test error on the labels

$$e_{\text{test}}(W) = \frac{1}{2} \mathbb{E}_{\mathbf{x}} \left( f^*(\mathbf{x}; W^*) - \hat{f}(\mathbf{x}; W) \right)^2 , \tag{5}$$

where the average is over a new test sample $\mathbf{x}$ with the same distribution as the training samples (notice that for the sake of the test error we do not add any label noise on the test label). The Bayes-optimal (BO) test error, i.e. the minimum test error achievable by any estimator (on average over the joint realization of the dataset and the target weights) is a natural baseline. In the high-dimensional limit (4), the BO test error has been characterized in [Maillard et al., 2024]. We will always consider the high-dimensional limit (4).

## 2 Main theorem

Our main technical result is the characterization of the properties of the global minima of the empirical loss (2) in the high dimensional limit in terms of training and test error.

**Theorem 1 (Asymptotics of ERM (2), informal)** *Consider the setting of Section 1 with $\kappa \geq 1$. Define $\tilde{\lambda} = \sqrt{\kappa} \lambda$ and $\mu_\delta^* = \mu^* \boxplus \mu_{\mathrm{s.c.},\delta}$, where $\boxplus$ is the free convolution and $\mu_{\mathrm{s.c.},\delta} = \sqrt{4\delta^2 - x^2}/(2\pi\delta^2)$ the semicircle distribution of radius $2\delta$ for $\delta > 0$. Call $(\bar{\delta}, \bar{\epsilon}) \in \mathbb{R}_+^2$ the unique solution of*

$$\begin{cases} 4\alpha\delta - \frac{\delta}{\epsilon} = \partial_1 J(\delta, \tilde{\lambda}\epsilon) \\ Q^* + \frac{\Delta}{2} + 2\alpha\delta^2 - \frac{\delta^2}{\epsilon} = (1 - \epsilon\tilde{\lambda}\partial_2)J(\delta, \tilde{\lambda}\epsilon) \end{cases} \quad \text{where } J(a,b) = \int_b^{+\infty} dx\, \mu_a^*(x)\, (x - b)^2 . \tag{6}$$

*Then, for all values of $\alpha, \kappa^*, \lambda > 0$, $\Delta \geq 0$ and $\kappa \geq 1$ any global minimum $\hat{W}$ of (2) satisfies*

$$\lim_{d \to \infty} \mathbb{E} e_{\text{test}}(\hat{W}) = 2\alpha\bar{\delta}^2 - \frac{\Delta}{2}, \qquad \lim_{d \to \infty} d^{-2} \mathbb{E} \mathcal{L}(\hat{W}) = \frac{\bar{\delta}^2}{4\bar{\epsilon}^2} - \frac{\tilde{\lambda}}{2} \partial_2 J(\delta, \tilde{\lambda}\epsilon). \tag{7}$$

*Moreover, if $\mu^*$ has compact support, then the empirical singular value density of $\hat{W}/\sqrt[4]{md}$ satisfies*

$$\lim_{d \to \infty} \mathbb{E} \frac{1}{d} \sum_{i=1}^d \delta(x - \sigma_i) = F_{\bar{\delta}}(\tilde{\lambda}\bar{\epsilon})\delta(x) + I(x > 0) \left[ 2x\mu_{\bar{\delta}}(x^2 + \tilde{\lambda}\bar{\epsilon}) \right] \tag{8}$$

*where $\{\sigma_i\}_{i=1}^d$ are the singular values of $\hat{W}/\sqrt[4]{md}$, $I$ is the indicator function and $F_\delta$ is the c.d.f. of $\mu_\delta^*$.*

*Sketch of the proof.* The proof, detailed in Appendix A, proceeds as follows. We use a reduction of a regularized version of the minimization (2) to the one of a positive semi-definite matrix estimation:

$$\hat{S} = \arg\min_{S \succeq 0} \tilde{\mathcal{L}}(S), \text{ with } \tilde{\mathcal{L}}(S) := \sum_{\mu=1}^n \text{Tr}\left[X(\mathbf{x}^\mu)(S - S^*)\right]^2 + \sqrt{md}\left(\lambda \text{Tr}(S) + \tau \|S\|_F^2\right) \tag{9}$$

where we defined $S^* = W^*(W^*)^T/\sqrt{m^* d}$ and $X(\mathbf{x}) = (\mathbf{x}\mathbf{x}^T - \mathbb{I}_d)/\sqrt{d}$, and where we have added a Frobenius norm penalty of amplitude $\tau$. Notice that (9) is strongly convex for $\lambda, \tau > 0$, so that

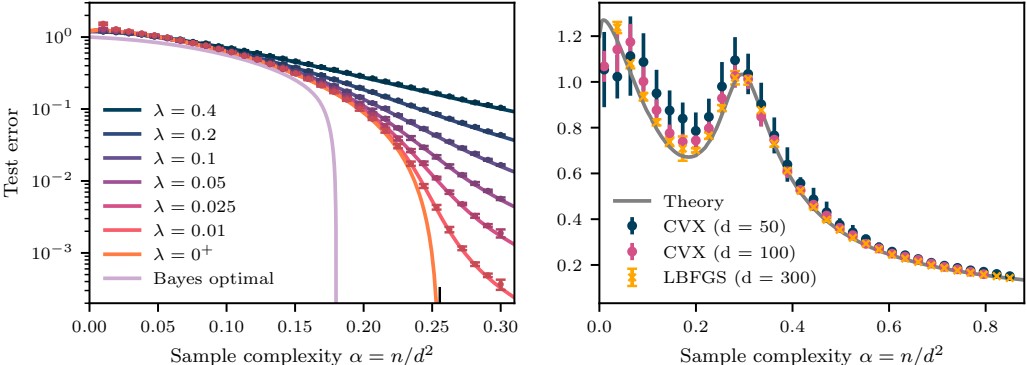

Figure 1: Left: Test error of simulations of vanilla GD (crosses, error bars are the standard deviation over 16 realizations of the target/training set at $d = 300$) compared with the results of Theorem 1 (lines) as a function of the number of samples $n = \alpha d^2$, noiseless case $\Delta = 0$, $\kappa^* = 0.2$. We observe a perfect match, particularly striking in the regime of small test error. The purple line is the Bayes-optimal performance [Maillard et al., 2024]. Right: Test error of simulations of GD run with LBFGS on (2) (yellow dots, $d = 300$) and of a convex solver run on the equivalent convex matrix problem (9) (blue dots $d = 50$, purple $d = 100$ dots), for $\Delta = 0.5$, $\kappa^* = 0.2$, and $\lambda = 0.02$ and as a function of the number of samples $n = \alpha d^2$. Error bars are the standard deviation over 16 realizations of the target/training set, compared with the result of Theorem 1 (gray line).

$\hat{S}$ is unique. Under the mapping $S(W) = WW^T/\sqrt{md}$, we have $\hat{S} = S(\hat{W})$ for any (Frobenius regularized) solution of (2), due to the uniqueness of $\hat{S}$. The second part is to use the Gaussian universality principle, that allows to replace each matrix $(\mathbf{x}\mathbf{x}^T - \mathbb{I}_d)/\sqrt{d}$ by a random Wigner matrix, following closely the steps of [Maillard et al., 2024, Xu et al., 2025]. This in turns reduces the problem to a rank-penalized matrix recovery problem with $\mathrm{GOE}(d)$ sensing matrices:

$$\hat{S} = \arg\min_{S \succeq 0} \tilde{\mathcal{L}}_G(S), \text{ with } \tilde{\mathcal{L}}(S) := \sum_{\mu=1}^{n} \mathrm{Tr}\left[Z^\mu (S - S^*)\right]^2 + \sqrt{md}\left(\lambda \mathrm{Tr}(S) + \tau \|S\|_F^2\right) \quad (10)$$

The problem can then be studied in various way. For instance with the heuristic, but powerful, replica method technique from statistical physics (see for instance [Maillard and Kunisky, 2024]). To provide a rigorous approach, we use instead a suitable Approximate Message Passing (AMP) algorithm with non-separable prior [Berthier et al., 2020, Gerbelot and Berthier, 2023], designed in such a way as its (unique) non-trivial fixed point is also the fixed point of projected gradient descent on (9) [Bayati et al., 2010, Montanari et al., 2012, Loureiro et al., 2021b], which by the convexity of (9) coincide with the unique $\hat{S}$. We then write the state evolution of this AMP algorithm which gives us all the relevant characteristics of the minimizer. Simple manipulations of the state evolution equations, in the limit $\tau \to 0$, lead to the characterization (6). $\qquad\square$

Theorem 1 provides an asymptotic result for train and test error, as well as a characterization of the singular values of the optimal neural network weights. It is of independent interest for the matrix compressed sensing problem (10) and extends directly to *any strictly convex matrix problem*.

A remarkable observation is that the $\ell_2$ regularization over the weights $W$ naturally translates to a nuclear norm regularization in the equivalent matrix problem (the convexification of the minimum rank regularization), naturally favoring model weights configurations with an effective lower width (i.e., implementable with fewer hidden units): a weight decay in $W$ thus implies a low nuclear norm of the matrix $S$.

Theorem 1 also implies that the properties of the global minima do not depend on the network width $m$, as long as $m \geq d$. This means that a neural network with width $\kappa \gg 1$ will achieve the same test error as a much narrower network with $\kappa = 1$, when trained on the same data, provided that the regularization strength is appropriately matched. Theorem 1 thus described both mildly and massively over-parametrized models since for very large $\kappa$ the number of learnable parameters will be not only much larger than the target function, but also can be much larger than the number of samples.

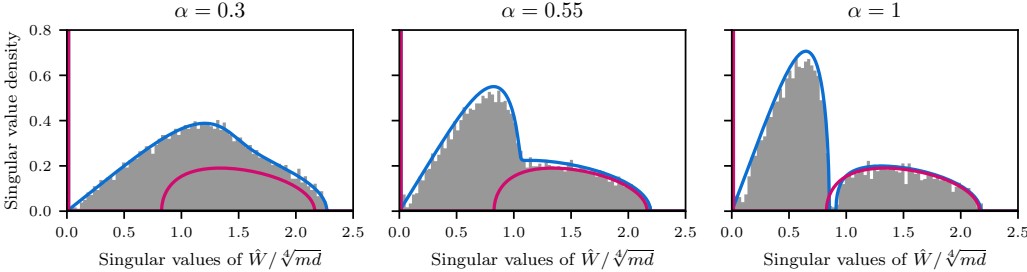

Figure 2: Spectra of the singular values of $\hat{W}/\sqrt[4]{md}$ for $\kappa^* = 0.2$, $\Delta = 0.5$, $\lambda = 0.02$ and several values of $\alpha$. The red line is the singular value density of the target $2x\mu^*(x^2)$, the blue line is the density predicted by (8). The histogram in gray is computed on the singular values of 16 runs of LBFGS on experiments with $d = 400$.

We remark that our results can be extended to the setting where the second layer weights of the student are binary $\pm 1$ with at least $d$ positive and $d$ negative weights, see Appendix G.

**Illustration of Theorem 1 and the behavior of gradient descent.** In Figure 1 we compare the asymptotic results of Theorem 1 with numerical experiments at finite size run directly on the equivalent convex loss (9) (using the solver CVXPY [Diamond and Boyd, 2016, Agrawal et al., 2018]). We notice that despite the theoretical results being valid in the high-dimensional limit, they are in excellent agreement with simulation at sizes as moderate as $d = 50$. Details on the numerical experiments are given in Appendix F.

The mapping onto a matrix problem implies immediately that Theorem 1 describes also the performance of gradient descent at convergence. Indeed, [Arjevani et al., 2025, Theorem 12] implies that if $m \geq d$, then the original minimization problem (2) has no local minima (in the language of [Arjevani et al., 2025], it has no spurious valleys, which implies in our case absence of local minima). We verify this claim experimentally at finite size by running gradient descent (GD)

$$w_{ki}^t = w_{ki}^{t-1} - \eta \nabla_{w_{ki}} \mathcal{L}(w) \tag{11}$$

initialized as $w_{ki}^{t=0} \sim \mathcal{N}(0, \zeta^2)$ independently, with $\hat{w}_{\mathrm{GD}} = w^T$, where $T = 10^4$ is a fixed stopping time (we check that by time $T$ GD has convincingly reached convergence) and $\eta$ a tuned learning rate (in Figure 1 right we used LBFGS for better convergence). We present our experiments in Figure 1. Again, we observe a very nice agreement, surprisingly precise in the region of small test error (notice the log-scale on the vertical axis in the left panel). We further test our results by comparing the empirical spectra with our theoretical prediction of equation (8) in Figure 2, observing a nice match.

**Relation with kernel ridge regression.** While the equivalent model (9) is linear in the quadratic feature space $\mathbf{x} \to x_i x_j$, out result departs significantly from the analysis of kernel ridge regression, random features regression and other linear methods studied in the literature (see e.g. [Gerace et al., 2020, Bordelon et al., 2020, Loureiro et al., 2021a, Mei and Montanari, 2022]). The crucial difference is in the regularization, which in (9) is a non-separable regularization *including the positive semi-definite constraint*, that takes into account the full spectral distribution of the target $\mu^*$, while in the literature only the case of $\ell_2$ regularization in feature space was usually studied explicitly (a separable regularization that takes into account only the second moment of the target $Q^*$, disregarding for example the low-width of the target function). Among the consequences of this, we find an interpolation threshold that, unexpectedly, depends on the structure of the target function (Result 1).

**Narrow students $\kappa^* \leq \kappa < 1$.** We remark that Theorem 1 applies as is also to models with $m < d$ for certain values of the parameters, i.e. for all choices of $\alpha, \kappa^*, \lambda, \Delta$ such that $m/d = \kappa > 1 - F_{\bar{\delta}}(\bar{\epsilon})$. This is a consequence of the fact that the global minima of (2) for $\kappa \geq 1$ have rank $F_{\bar{\delta}}(\bar{\epsilon})$ (see (8)), and thus are also global minima of the (2) for $\kappa < 1$ under the above condition. Lower than this value of $\kappa$, the rank constraint imposed by the structure of the student network affects the global minima of the unconstrained (9), leading to non-convexity and to the failure of Theorem 1.

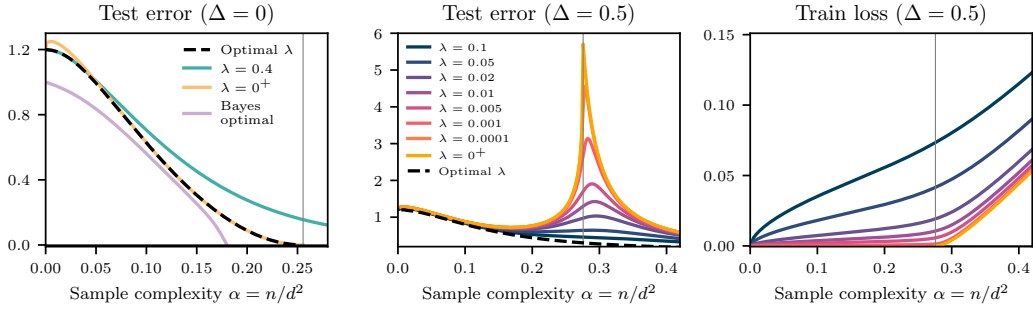

Figure 3: (Left) The test error of any global minimum of (2) (Theorem 1) in the noiseless case $\Delta = 0$, $\kappa^* = 0.2$ for finite regularization $\lambda = 0.4$ (blue line), in the limit $\lambda \to 0^+$ (yellow line) and for optimal regularization (dashed line). We compare with the Bayes-optimal performance [Maillard et al., 2024] (purple line), and highlight the strong recovery threshold (vertical gray line, see Corollary 1). (Center, Right) The test and train loss (2) in the noisy case $\Delta = 0.5$, $\kappa^* = 0.2$ for several values of the regularization $\lambda$ (solid lines), $\lambda \to 0^+$ (yellow line) and for optimal regularization (dashed line). We highlight the region of sample ratio $\alpha$ where non-regularized training loss goes to zero (before the vertical grey line, from Result 1), which coincides with the development of a cusp in the test error as $\lambda$ decreases.

## 3 Main results and consequences of Theorem 1

In this section we focus specifically on the Marchenko-Pastur target case (see Section 1, details on the computation of $\mu_\delta^*$ in Result 1 are given in Appendix E). We believe that our results generalize (qualitatively for the learning curves, and as is for the thresholds and the low-rank target limit) to any target such that $\mu^*$ has a finite spectral gap, i.e. such that $\lambda_{\min} = \min\{x \in \text{supp}(\mu^*) | x > 0\}$ satisfies $\lambda_{\min} > 0$, with mass in zero equal to $\max(0, 1 - \kappa^*)$ (i.e. rank of the associated target matrix $\lfloor \min(\kappa^*, 1)d \rfloor$) and second moment $Q^* = 1 + \kappa^*$.

**Learning curves.** In Figure 3 we show the test and train error of any global minimum of the loss function (2) given by Theorem 1, both the noiseless ($\Delta = 0$) and noisy ($\Delta = 0.5$) scenarios. Details on the numerical experiments are given in Appendix E.

In the noiseless case (Figure 3 left), we plot the test error as functions of the sample ratio $\alpha$ for a finite regularization value ($\lambda = 0.4$, blue line), in the limit of vanishing regularization ($\lambda \to 0^+$, yellow line), and for the optimal choice of regularization (dashed line). These are contrasted with the Bayes-optimal performance as derived in [Maillard et al., 2024] (purple line), clearly illustrating the effect of regularization on the perfect recovery threshold, marked by the vertical gray line as per Corollary 1.

In the noisy case (Figure 3 center and right), we show both the test and train losses as functions of the sample ratio $\alpha$ for multiple values of $\lambda$ (solid lines), including the minimum regularization interpolator limit ($\lambda \to 0^+$, yellow line) and the optimally regularized scenario (dashed line). Notably, we highlight the specific region of $\alpha$ where the non-regularized training loss reaches zero (vertical gray line, as per Result 1), which coincides with the emergence of a cusp – the double descent peak [Belkin et al., 2019, Nakkiran et al., 2021] – in the test error as $\lambda$ decreases.

**Interpolation threshold.** We define the interpolation threshold $\alpha_{\text{inter}}(\kappa^*, \Delta)$ as the value of the sample ratio at which the set of PSD matrices $S$ for which the loss (9) is zero for $\lambda = 0$ (we call those the interpolators of the training dataset) shrinks to a single point. For $\alpha > \alpha_{\text{inter}}(\kappa^*, \Delta)$, (9) thus has a single global minimum with positive training loss if $\Delta > 0$, or with both zero training and test loss if $\Delta = 0$. For $\alpha < \alpha_{\text{inter}}(\kappa^*, \Delta)$ instead, the non-regularized $\lambda = 0$ loss admits many global minima (all PSD interpolators), while the limit $\lambda \to 0^+$ has still a single global minimum with zero training loss, and minimum value of the regularization norm. While we stated Theorem 1 for $\lambda > 0$ it also holds, at least at an heuristic level, for $\alpha > \alpha_{\text{inter}}(\kappa^*, \Delta)$ and $\lambda = 0$ by directly plugging $\lambda = 0$ in (6). The theorem *does not hold* for $\lambda = 0$ and $\alpha < \alpha_{\text{inter}}(\kappa^*, \Delta)$ while it can be adapted to the limit $\lambda \to 0^+$ by a careful rescaling, which we comment upon in Appendix B. A

direct consequence of Theorem 1 is a theoretical characterization of the position of the interpolation threshold $\alpha_{\text{inter}}(\kappa^*, \Delta)$.

**Result 1 (Interpolation threshold)** *Consider the setting of Section 1 for a Marchenko-Pastur target. Then, the interpolation threshold $\alpha_{\text{inter}}(\kappa^*, \Delta)$ satisfies*

$$\alpha_{\text{inter}}(\kappa^*, \Delta) = \frac{1}{4\bar{\delta}} J^{(1,0)}\left(\bar{\delta}, 0\right) + o_d(1) \quad where \quad Q^* + \frac{\Delta}{2} = J\left(\bar{\delta}, 0\right) - \frac{\bar{\delta}}{2} \partial_1 J\left(\bar{\delta}, 0\right) , \quad (12)$$

*where $\partial_1$ denotes derivative w.r.t. the first argument, and $J$ was defined in (6). Additionally, we have*

$$\lim_{\Delta \to 0^+} \alpha_{\text{inter}}(\kappa^*, \Delta) = \frac{1}{4} \begin{cases} 1 + 2\kappa^* - \kappa^{*2} & \text{if } 0 < \kappa^* < 1 \\ 2 & \text{if } \kappa^* \geq 1 \end{cases} , \quad \lim_{\Delta \to +\infty} \alpha_{\text{inter}}(\kappa^*, \Delta) = \frac{1}{4} . \quad (13)$$

Result 1 is derived by considering the solution of the minimization problem (6) and setting $\lambda = 0$ directly at large $\alpha$ – where there is a unique minimum of (9) – and then looking for the smallest value of $\alpha$ consistent with a unique minimum assumption. Equivalently, this is when the asymptotics training loss reach zero (see Appendix C). We plot (13) in Figure 4. As discussed above, the interpolation threshold does not depend on the width of the student.

It is interesting to contrast our results with the *ellipsoid-fitting problem*, where one seeks a PSD matrix $S$ such that a dataset of points $\{\mathbf{x}^\mu\}_{\mu=1}^n$ lies on the surface of the associated ellipsoid. This problem, recently studied in a high-dimensional limit similar to (4) [Maillard and Kunisky, 2024, Maillard and Bandeira, 2023], predicts that fitting is possible as long as $\alpha < 1/4$. We recover this same threshold, $\alpha_{\text{inter}} = 1/4$, in two extreme cases: when noise dominates ($\Delta \to +\infty$) or when the target is vanishingly small ($\Delta = 0, \kappa^* \to 0^+$). In all other scenarios, however, the interpolation threshold is strictly larger, reflecting the structural advantage of the data over random labeling. Unlike linear ridge regression or kernel methods, the interpolation threshold here is nontrivial and intricately linked to data structure. Notably, it is not simply determined by the ratio of samples to parameters. Despite the effective number of parameters scaling as $\mathcal{O}(d^2/2)$, interpolation can occur well before this count if the target function is sufficiently narrow ($0 < \kappa^* < 1$). This effect arises from the PSD constraint in the equivalent matrix problem (9), marking a fundamental departure from classic kernel theory.

We note that special cases this threshold has been considered in the literature, for $\kappa^* \geq 1$ and $\Delta = 0$ in [Gamarnik et al., 2019], and our result agrees with these works. An analogue of this threshold (still for $\Delta = 0$) has been considered in [Sarao Mannelli et al., 2020] for the case of low-width target $m^* = 1$, where the interpolation for the model with $\kappa \geq 1$ happens at $n = 2d$. The authors conjecture without a solid theoretical justification that for generic $m^*$ the interpolation happens at $n = d(m^* + 1) - m^*(m^* + 1)/2$, which, however, is not compatible with our results. We conclude that their conjecture is incorrect in the regime $m^* = \mathcal{O}_d(d)$, as it predicts interpolation for values lower than the rigorous $1/4$ random labels threshold [Maillard and Bandeira, 2023, Maillard and Kunisky, 2024] as well.

**Strong recovery threshold.** A different phenomenon can be studied in the noiseless case $\Delta = 0$ and $\lambda \to 0^+$, i.e. what we call the *strong recovery*: the value of the sample ratio $\alpha_{\text{strong}}(\kappa^*) = n_{\text{strong}}/d^2$ such that, for $\alpha > \alpha_{\text{strong}}$ the *test error* of the global minima of (2) is zero, and for $\alpha < \alpha_{\text{strong}}$ is strictly positive at $\Delta = 0$. Notice the difference with the previous paragraph, where the interpolation peak was concerning the training error vanishing in the $\Delta > 0$ case, while we here consider the perfect test error at $\Delta = 0$. This limit was studied in the context of matrix model (9) in the minimal-norm interpolation setting [Donoho et al., 2013, Amelunxen et al., 2014, Oymak and Hassibi, 2016]. Here we re-derive their result independently as a consequence of Theorem 1.

**Corollary 1 (Strong recovery threshold)** *Consider the setting of Section 1 for Marchenko-Pastur targets with $\Delta = 0$, $\lambda \to 0^+$ and $\kappa \geq 1$. Define for $x \in [-2, 2]$ the incomplete moments of the semicircle distribution*

$$M_{\text{s.c.}}^{(k)}(x) = \int_{-2}^{x} dx\, \mu_{\text{s.c.}}(x)\, x^k = \frac{1}{2\pi} \int_{-2}^{x} dx\, \sqrt{4 - x^2}\, x^k , \quad (14)$$

*and call $\bar{c}$ the solution of the equation*

$$M_{\text{s.c.}}^{(1)}(c) - c M_{\text{s.c.}}^{(0)}(c) + \frac{c}{1 - \kappa^*} = 0 , \quad (15)$$

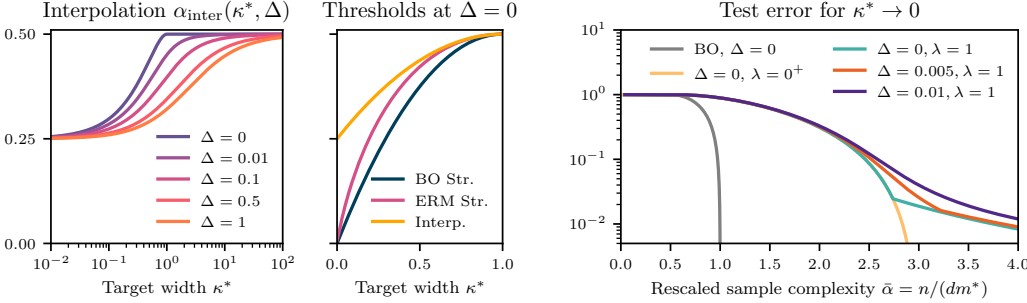

Figure 4: (Left) Interpolation threshold $\alpha_{\mathrm{inter}}(\kappa^*, \Delta)$ as a function of $\kappa^*$ for several values of label noise $\Delta$ (Result 1). Notice the convergence to the $1/4$ random-label-fitting threshold for very narrow targets $\kappa^* \ll 1$ and large label noise $\Delta \gg 1$. (Center) Comparison between interpolation threshold (Result 1, $\Delta = 0$) and strong recovery threshold (Corollary 1) of the global minima of (2), with the BO strong recovery threshold [Maillard et al., 2024]. Minimal regularization interpolators of (2) reach perfect recovery well before the interpolator set shrinks to a singleton on the target weights (the effect is more pronounced for very small ranks of the target function $\kappa^* \ll 1$. (Right) The test error of any global minimum of (2) in the limit $\kappa^* \to 0$ (Result 2) for several values of regularization $\lambda = \bar{\lambda}/\sqrt{\kappa^*}$ and label noise $\Delta$, compared with the Bayes-optimal [Maillard et al., 2024].

*for $0 < \kappa^* < 1$. Then,*

$$\alpha_{\mathrm{strong}} = \begin{cases} \frac{1}{2} - \frac{1}{2}(1 - \kappa^*)^2 \left( M_{\mathrm{s.c.}}^{(2)}(c) - c M_{\mathrm{s.c.}}^{(1)}(c) \right) & \text{if} \quad 0 < \kappa^* < 1 \\ \frac{1}{2} & \text{if} \quad \kappa^* \geq 1 \end{cases} + o_d(1). \tag{16}$$

Informally, the strong recovery threshold presented in Result 1 is derived by expanding (6) for $\delta \to 0^+$ within an appropriate scaling ansatz, as the test error is proportional to $\delta^2$ (Theorem 1, noiseless case $\Delta = 0$), and finding the value $\alpha = \alpha_{\mathrm{strong}}$ such that the expansion is consistent. We provide all the details in Appendix C. We plot (16) in Figure 4, comparing with the strong recovery threshold of the BO estimator [Maillard et al., 2024] and with the interpolation threshold Result 1 for $\Delta = 0$.

**Small target rank limit.** Finally, we consider the small target width limit $\kappa^* \to 0^+$ for $\kappa$ bounded away from zero. This is an example of large over-parametrization, as the ratio between student and target widths $m/m^*$ diverges. In the limit $\kappa^* \to 0$, i.e. $m^* \ll d$, we define the rescaled sample ratio $\bar{\alpha} = \alpha/\kappa^* = n/(m^* d)$. The natural baseline to compare this limit against is given by the analogue limit for the test error of the BO estimator given in [Maillard et al., 2024].

**Result 2 (Test error in the $\kappa^* \to 0$ limit)** *Consider the setting of Section 1. Consider the limit $\kappa^* \to 0^+$ for any fixed $\kappa > 0$, $\bar{\alpha} = \alpha/\kappa^*$ and $\bar{\lambda} = \lambda\sqrt{\kappa^*}$. Then, if $\Delta = 0$ and $\bar{\lambda} \to 0^+$ the test error satisfies*

$$\lim_{\kappa^* \to 0^+} \lim_{d \to \infty} \mathbb{E}\, e_{\mathrm{test}}(\hat{S}) = \begin{cases} 1 & \text{if} \quad 0 < \bar{\alpha} \leq 1/2, \\ \frac{2}{9}\bar{\alpha}(4 - \sqrt{6\bar{\alpha} - 2})^2 & \text{if} \quad 1/2 < \bar{\alpha} \leq 3 \\ 0 & \text{if} \quad \bar{\alpha} \geq 3 \end{cases} \tag{17}$$

*In Appendix D we provide analogous expressions, albeit less explicit, for generic regularization $\bar{\lambda} > 0$ and noise $\Delta \geq 0$. We just state that for all $\bar{\lambda} > 0$ and $\Delta \geq 0$, and for $\alpha < \bar{\alpha}_{\mathrm{weak}}(\bar{\lambda}, \Delta)$ the test error equals one, where*

$$\bar{\alpha}_{\mathrm{weak}}(\bar{\lambda}, \Delta) = \max\left( \frac{1 + \Delta/2}{2}, \frac{\bar{\lambda} - 2(1 + \Delta/2)}{4} \right), \tag{18}$$

*and that for all $\bar{\lambda} > 0$ and $\Delta > 0$ and for $\bar{\alpha} \to +\infty$ he have*

$$\lim_{\kappa^* \to 0^+} \lim_{d \to \infty} \mathbb{E}\, e_{\mathrm{test}}(\hat{S}) = \frac{3\Delta}{2\bar{\alpha}}(1 + o_{\bar{\alpha}}(1)). \tag{19}$$

Result 2 provides a closed-form expression for the test error in the $\kappa \to 0$ limit. It is derived by a direct expansion of (6) at small $\kappa$, and it involves a BBP-like phase transition [Baik et al., 2005] in the spectrum of $\hat{W}$. We notice that Result 2 holds for any extensive student width $\kappa > 0$ due to the fact that the rank of the global minima of (2) is vanishing as $\mathcal{O}(\kappa^*)$.

Figure 4 right shows the test error in the $\kappa^* \to 0$ limit for a selection of parameters. We see that the solution at $\lambda, \Delta \neq 0$ as a non-trivial behavior, tracking closely but not exactly the $\lambda \to 0^+$, $\Delta \neq 0$ curve until an angular point is reached. We observe that for $\bar{\alpha} < \bar{\alpha}_{\text{weak}}(\bar{\lambda}, \Delta)$, the empirical risk minimizer achieves BO performance, even though that is given by the trivial prior-average estimator. Moreover, for $\Delta = 0$ and $\bar{\lambda} \to 0^+$, the empirical risk minimizer achieves strong recovery for $\bar{\alpha} = 3$ in accordance with Corollary 1. Finally, we remark that for fixed $\bar{\lambda} > 0$ and $\Delta > 0$, the test error goes to zero as $1/\alpha$, which is the Bayes-optimal rate.

## 4   Discussion and limitations

In this paper, we presented a closed-form asymptotic characterization of learning a one hidden-layer quadratic neural network target with an over-parametrized architecture, in the challenging high-dimensional regime of extensive widths $m^* = \mathcal{O}(d)$ target functions and quadratically-many samples $n = \mathcal{O}(d^2)$. We presented the asymptotic learning curves and values of relevant thresholds, as well as numerical experiments (including gradient descent algorithms) illustrating that our theory applied also accurately to very moderate sizes.

The main limitations of our setting are the restriction to random input data (which can be relaxed to a certain extent, see [Xu et al., 2025] for a theoretical direction hinging on Gaussian universality, or Loureiro et al. [2021b] for an empirical direction modeling real data by matching to Gaussian mixtures with appropriate first and second moments), the choice of quadratic activation, the single hidden-layer architecture and the requirement that the width of the target function satisfies $m > d$ (imposing no non-convex rank constraint in (9)). All such elements present highly non-trivial technical challenges, and we hope our work will spark progress in these directions.

## Acknowledgments and Disclosure of Funding

We thank Yatin Dandi, Cedric Gerbelot, Antoine Maillard, Simon Martin, Yizhou Xu and Nathan Srebro for insightful discussions. This work was supported by the Swiss National Science Foundation under grants SNSF SMArtNet (grant number 212049), SNSF OperaGOST (grant number 200390), and SNSF DSGIANGO (grant number 225837).

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

# A Proof of Theorem 1

In this section we discuss theorem 1 for the minima of

$$\hat{W} = \arg\min \mathcal{L}(W), \quad \text{where} \quad \mathcal{L}(W) := \sum_{\mu=1}^{n} \left( y^\mu - \hat{f}(\mathbf{x}^\mu; W) \right)^2 + \lambda \|W\|_F^2 + \tau \|WW^T\|_F^2 \,.$$
(20)

which, due to the added regularization, is strongly convex for $\tau, \lambda > 0$ and hence it admits a unique global minimum. Note that while we concentrate on this case, the computation goes through for any (strongly) convex denoiser.

The proof of Theorem 1 is an application of standard approaches in high-dimensional statistics, and goes as follows:

1. In Section A.1, we show how to map the original learning problem into an equivalent matrix estimation problem. This, in turn, can be reformulated as a vector Generalized Linear Model (GLM) with a non-separable regularization.

2. In Section A.2, we use Gaussian universality and show that this model is asymptotically equivalent to a Gaussian sensing problem, leveraging recent results on universality in high-dimensional estimation.

3. The resulting Gaussian model can then be analyzed using the Generalized Approximate Message Passing (GAMP) algorithm. We briefly recall the relevant properties of GAMP in Section A.3.

   To apply GAMP rigorously, we exploit the fact that its fixed points coincide with the solutions of the associated convex optimization problem (Section A.3.1). These fixed points are characterized by a set of deterministic equations known as state evolution (Section A.3.2). With appropriate initialization (see Corollary 2), GAMP converges to the correct fixed point (Theorem 5), which corresponds to the global minimizer of our original problem. Hence, the fixed points of the associated state evolution equations describe the properties of the global minima of (1). Notice that this discussion is generic (i.e. allows for generic convex losses and regularizations), as long as the global minimum of the associated matrix problem is unique.

4. Finally, in Appendix A.4, we describe the state evolution equations that precisely characterizes the asymptotic behavior of the minimizer of the empirical risk (2), which is precisely equivalent to the results given in Theorem 1 in the limit $\tau \to 0^+$.

## A.1 Mapping to a matrix model

The mapping of the data part of the loss is straightforward. For the $\ell_2$ regularization on the network weights we have

$$\sum_{k=1}^{m} \|\mathbf{w}_k\|_2^2 = \sum_{i,k=1}^{d,m} W_{ki}^2 = \sqrt{md} \sum_{i,k=1}^{d,m} \frac{W_{ki}^2}{\sqrt{md}} = \sqrt{md} \sum_{i=1}^{d} S(W)_{ii} = \sqrt{md} \, \text{Tr}(S(W)) \,.$$
(21)

The same goes for the part proportional to $\tau$ in (20), that become the Frobenius norm of $\hat{S}$. This leads to the equivalent problem:

$$\hat{S} = \arg\min_{S \succeq 0} \tilde{\mathcal{L}}(S), \;\; \text{with} \;\; \tilde{\mathcal{L}}(S) := \sum_{\mu=1}^{n} \text{Tr} \left[ \frac{\mathbf{x}_\mu \mathbf{x}_\mu^T - \mathbb{I}_d}{\sqrt{d}} \left( S - S^* \right) \right]^2 + \sqrt{md} \left( \lambda \, \text{Tr}(S) + \tau \|S\|_F^2 \right) \,.$$
(22)

## A.2 Data universality

The universality of the minimal error (or "ground state energy" in the physics jargon) follows directly from a rich line of work on universality of empirical risk minimization [Goldt et al., 2022, Hu and Lu, 2022, Montanari and Saeed, 2022, Maillard and Bandeira, 2023, Martin et al., 2024b, Xu et al., 2025]. These results establish that, under broad conditions, the asymptotic behavior of the test and

training errors becomes equivalent to those with a Gaussian sensing matrix. Given the maturity of this theory, we will refer directly to these foundational works and provide a concise sketch of the main arguments for completeness.

The universality follows directly from Proposition 2.1 in [Maillard and Bandeira, 2023], who derived it in the related context of ellipsoid fitting. This was adapted to the "planted" case in [Xu et al., 2025, Section 3.1, Lemma 3.3]. The proof strategy is to consider an interpolating model where each data matrix is given by $U(t) = \cos(t)(\mathbf{x}_\mu \mathbf{x}_\mu^T - \mathbb{I}_d)/\sqrt{d} + \sin(t)G_\mu$ in $t \in [0, 2\pi]$, from the original model at $t = 0$ to the one with a $\mathrm{GOE}(d)$ random matrix $G_\mu$ at $t = \pi/2$, and to show that the "time" $t$ does not change the expected loss. We refer to these works (and to [Montanari and Saeed, 2022]) for the detailed proofs, and directly state the universality result:

**Theorem 2 (Gaussian Universality of the loss, from [Maillard et al., 2024, Xu et al., 2025])**
*Let $\psi : \mathbb{R} \to \mathbb{R}$ be a bounded differentiable functions with bounded derivative, then for any finite $\alpha = n/d^2$, the minimum of the training loss $\mathrm{GS}(\{\Phi_\mu\}_{\mu=1}^n) := \arg\min_S \tilde{\mathcal{L}}(S)(\{\Phi_\mu\}_{\mu=1}^n)$ (9) is universal with respect to the training data $\Phi_\mu$ in the sense that for any such $\psi$, then*

$$\lim_{d\to\infty} \left| \mathbb{E}_{\{\mathbf{x}_\mu\}} \psi \left[ GS_d \left( \left\{ \frac{\mathbf{x}_\mu \mathbf{x}_\mu^T - \mathbb{I}_d}{\sqrt{d}} \right\}_{\mu=1}^n \right) \right] - \mathbb{E}_{\{G_\mu\} \overset{i.i.d.}{\sim} GOE(d)} \psi \left[ GS_d(\{G_\mu\}_{\mu=1}^n) \right] \right| = 0. \quad (23)$$

*if the matrices $(\{\mathbf{x}_\mu \mathbf{x}_\mu^T - \mathbb{I}_d\})/\sqrt{d}$ respects the so-called one-dimensional CLT property [Xu et al., 2025, Assumption 2.2] and if target weights $S^*$ and the regularization term of the loss $r(S)$ satisfy [Xu et al., 2025, Assumption 2.1].*

The required one-dimensional CLT [Goldt et al., 2022, Montanari and Saeed, 2022, Dandi et al., 2023] property is a point-wise normality of the projection of the operator, proven in the case of Gaussian $\mathbf{x}^\mu$ in [Maillard and Bandeira, 2023, Lemma 4.8].

The universality argument can be readily adapted to other quantity as well, in order to show that not only the loss, but also other observables such as the overlaps are universal. We will not repeat the full proof, and directly use Theorem 3 in [Dandi et al., 2023] that can be applied *mutatis mutandis*[1]:

**Theorem 3 (Universality of the overlaps [Dandi et al., 2023], informal)** *For any bounded-Lipchitz function $\Psi : \mathbb{R}^{d^2} \times \mathbb{R}^{d^2} \to \mathbb{R}$, we have:*

$$\mathbb{E}\left[ \Psi\left(\hat{S}, S^*\right) \right] \xrightarrow[\substack{n,d\to+\infty \\ n/d^2 = \alpha > 0}]{} \mathbb{E}_{\mathcal{G}}\left[ \Psi\left(\hat{S}, S^*\right) \right]$$

*where $\hat{S} = \arg\min_S \tilde{\mathcal{L}}(S)$, and $S^*$ determines the target function.*

This directly can be applied to the overlap $\mathrm{Tr}[\hat{S}^T \hat{S}]$ and $\mathrm{Tr}[\hat{S}^T S^*]$, which are thus universal, as well as to the test error. Notice indeed that

$$e_{\mathrm{test}}(\hat{w}) = \frac{1}{2}\mathbb{E}_x \left(f_{m^*}(x; w^*) - f_m(x; \hat{w})\right)^2 = \frac{1}{d}||S(w) - S(w^*)||_F^2, \quad (24)$$

leading to an expression for the test error as a function of the overlaps. Finally, note that one can apply the same argument to show the universality of any finite $k \in \mathbb{Z}_+$ moment $d^{-1}\mathrm{Tr}[\hat{S}^k]$ (hinging crucially on the PSD constraint to claim that all such perturbations are convex). With mild regularity assumptions (for e.g. compact support, which is satisfied by the Marchenko-Pastur target case studied in detail in the main text), this implies the universality of the spectral distribution of $\hat{S}$ as well.

---

[1]The argument is a classic approach of large deviation, or Legendre transform. This is done by applying the universality to a *perturbed* version of the loss, where —to use the parlance of physics— a source term, or a regularization, such as $\lambda_S \mathrm{Tr}[\hat{S}^T S^*]$, is added. Due to the convexity of the loss in $\lambda_S$ the derivative with respect to $\lambda_S$ must converges to the same exact quantity in both the original and the equivalent model, since they converge point-wise to the same limit.

## A.3 Generalized Approximate Message Passing algorithm and state evolution equations

We review here how to use GAMP to solve asymptotics convex optimization problems. Later, in section A.4.1 we show indeed that the present PSD matrix optimization problem can be written as a vector one, with a non-separable denoiser.

We will work in a vector setting, where one has sensing matrices $A \in \mathbb{R}^{N \times D} \sim \mathcal{N}(0, 1/D)$ (i.e. $N$ samples in $D$ dimension) with rows $\boldsymbol{a}^\mu$.

Notice that in this scaling $||\boldsymbol{u}||^2$ should be always of order $\mathcal{O}(D)$.

The generic form of the "rectangular" GAMP algorithm reads [Javanmard and Montanari, 2013]:

$$
\begin{aligned}
\boldsymbol{u}^{t+1} &= A^T g_t(\boldsymbol{v}^t) + d_t e_t(\boldsymbol{u}^t), \\
\boldsymbol{v}^t &= A e_t(\boldsymbol{u}^t) - b_t g_{t-1}(\boldsymbol{v}^{t-1}),
\end{aligned}
\tag{25}
$$

with initialization $\boldsymbol{u}^{t=0}$ and the convention that $g_{-1}(\cdot) = 0$. Here $\boldsymbol{v} \in \mathbb{R}^N$, $\boldsymbol{u} \in \mathbb{R}^D$, $g : \mathbb{R}^N \to \mathbb{R}^N$ and $e : \mathbb{R}^D \to \mathbb{R}^D$.

The terms $b_t, d_t$ are usually called *Onsager's reaction terms* [Mézard et al., 1987], and are tuned in such a way as to erase time-to-time correlations between the iterates $(\boldsymbol{u}^t, \boldsymbol{v}^t)$ and the sensing matrix $A$, so that from a statistical point of view of $(\boldsymbol{u}^t, \boldsymbol{v}^t)$ it is as if the matrix $A$ was resampled independently at each time-step (conditioned on the previous iterates $\{(\boldsymbol{u}^s, \boldsymbol{v}^s)\}_{s=1}^{t-1}$). One has

$$
d_t = -\frac{1}{D} \nabla \cdot g_t(\boldsymbol{v}^t) \quad \text{and} \quad b_t = \frac{1}{D} \nabla \cdot e_t(\boldsymbol{u}^t),
\tag{26}
$$

where $\nabla \cdot f = \sum_{i=1}^d \partial_i f_i$ denotes the divergence of a function $f : \mathbb{R}^d \to \mathbb{R}^d$. We will see later that in high-dimension $D \gg 1$, $N = \mathcal{O}(D)$, the Onsager's term concentrate and do not depend on the iterates $\boldsymbol{u}^t$ and $\boldsymbol{v}^t$ anymore.

### A.3.1 GAMP for convex optimization

Using AMP for studying the minimum of a loss is now a classical approach that has been used in many context, e.g. [Bayati et al., 2010, Montanari et al., 2012, Rangan et al., 2016, Loureiro et al., 2021b, Vilucchio et al., 2025]. This is also discussed in detail the pedagogical review [Feng et al., 2022, Section 4.4] and in the lecture notes [Krzakala and Zdeborová, 2024, Section 12.7.4].

The argument goes as follows: consider the convex optimization problem

$$
\arg\min_{\boldsymbol{u} \in \mathcal{C}} \sum_{\mu=1}^N \ell(\boldsymbol{u} \cdot \boldsymbol{a}, \boldsymbol{u}^* \cdot \boldsymbol{a}) + R(\boldsymbol{u}),
\tag{27}
$$

with $\ell, R$ convex in their first argument, and $\mathcal{C} \subseteq \mathbb{R}^D$ a convex set. Define the functions

$$
\begin{aligned}
\bar{g}(r, s, \bar{b}) &= \arg\min_{z \in \mathbb{R}} \left[ \ell(z, s) + \frac{1}{2\bar{b}}(z - r)^2 \right] \quad \text{and} \quad g(r, s, \bar{b}) = \frac{\bar{g}(r, s) - r}{\bar{b}}, \\
e(\boldsymbol{r}, \bar{d}) &= \arg\min_{\boldsymbol{z} \in \mathcal{C}} \left[ R(\boldsymbol{z}) + \frac{\bar{d}}{2} ||\boldsymbol{z} - \bar{d}^{-1} \boldsymbol{r}||^2 \right].
\end{aligned}
\tag{28}
$$

Consider the GAMP algorithm

$$
\begin{aligned}
\boldsymbol{u}^{t+1} &= A^T g(\boldsymbol{v}^t, \boldsymbol{y}, b^t) + d_t e(\boldsymbol{u}^t, d^t), \\
\boldsymbol{v}^t &= A e(\boldsymbol{u}^t, d^t) - b_t g(\boldsymbol{v}^{t-1}, \boldsymbol{y}, b^{t-1}),
\end{aligned}
\tag{29}
$$

where $g$ is applied component-wise to $\boldsymbol{v}$ and $\boldsymbol{y}$. Then, [Feng et al., 2022, Proposition 4.4] guarantees that *fixed points of* (29) *are solutions of* (27) (by a minor adaptation, as they consider separable regularization, but all their arguments generalize directly).

As long as $\ell, R$ are convex in their first argument, state evolution follows automatically for this choices of $g$ and $e$.

Additionally, we remark that if the loss is *strictly convex*, then the optimization problem (27) has only a single global minimum, so that AMP will have a single fixed point coinciding with this global minimum.

### A.3.2 State evolution

Thanks to the conditional Gaussianity discussed in Appendix A.3, one can track the statistics of the iterated $\boldsymbol{u}, \boldsymbol{v}$ of GAMP through a set of so-called *state evolution equations*. The main difference with respect to previous approaches in our case is that the denoising function $e$ is not separable. However, [Berthier et al., 2020, Gerbelot and Berthier, 2023] guarantees that the state evolution that allows to track the performance of AMP remains corrects (under regularity assumptions that are automatically satisfied by $e$ and $g$ being proximal operators).

The state evolution associated to the GAMP

$$
\begin{aligned}
\boldsymbol{u}^{t+1} &= A^T g_t(\boldsymbol{v}^t, \boldsymbol{y}, b^t) + d_t e_t(\boldsymbol{u}^t, d^t), \\
\boldsymbol{v}^t &= A e_t(\boldsymbol{u}^t, d^t) - b_t g_{t-1}(\boldsymbol{v}^{t-1}, \boldsymbol{y}, b^{t-1})
\end{aligned}
\tag{30}
$$

(where again here we consider $g$ applied component-wise to $\boldsymbol{v}$ and $\boldsymbol{y}$) is derived by considering that $y^\mu = \phi(\boldsymbol{u}^* \cdot \boldsymbol{a}^\mu)$ where $\phi$ is a possibly random non-linearity, and taking into account the evolution of the iterates $\boldsymbol{u}$ and $\boldsymbol{v}$ separately in the parallel and orthogonal directions to $\boldsymbol{u}^*$ (and similarly with $\boldsymbol{v}$ and $\boldsymbol{v}^* = A^T \boldsymbol{u}^*$). Using [Berthier et al., 2020, Gerbelot and Berthier, 2023] we have:

**Theorem 4 (Theorem 1 in [Berthier et al., 2020], informal)** *Define*

$$
\begin{cases}
m_u^{t+1} &= \frac{N}{D} \mathbb{E}_u^t[\partial_{z^*} g(\omega, \phi(z^*), b^t)] \\
q_u^{t+1} &= \frac{N}{D} \mathbb{E}_u^t[(g(\omega, \phi(z^*), b^t)^2] \\
d^{t+1} &= -\frac{N}{D} \mathbb{E}_u^t \partial_\omega g(\omega, \phi(z^*), b^t) \\
m_v^t &= \frac{1}{D} \mathbb{E}_v^t \left[ \left\langle \mathbf{u}^* ; e(\sqrt{q_u^t} \boldsymbol{z} + m_u^t \mathbf{u}^*, d^t) \right\rangle \right] \\
q_v^t &= \frac{1}{D} \mathbb{E}_v^t \left[ \left\langle e(\sqrt{q_u^t} \boldsymbol{z} + m_u^t \mathbf{u}^*, d^t) ; e(\sqrt{q_u^t} \boldsymbol{z} + m_u^t \mathbf{u}^*, d^t) \right\rangle \right] \\
b^t &= \frac{1}{D} \mathbb{E}_v^t \nabla \cdot e(\sqrt{q_u^t} \boldsymbol{z} + m_u^t \mathbf{u}^*, d^t)
\end{cases}
\tag{31}
$$

*where the average $\mathbb{E}_u^t$ is over the randomness in $\phi$ as well as over $(\omega, z^*) \in \mathbb{R}^2$ distributed as*

$$
\begin{bmatrix} z^* \\ \omega \end{bmatrix} \sim \mathcal{N}\left( \begin{bmatrix} 0 \\ 0 \end{bmatrix} ; \begin{bmatrix} ||\boldsymbol{u}^*||^2/D & m^t \\ m^t & q^t \end{bmatrix} \right),
\tag{32}
$$

*and $\mathbb{E}_v^t$ is over $\boldsymbol{z}$ i.i.d. Gaussian $\mathcal{N}(0, \mathbb{I}_D)$.*

*Then the AMP vectors $\boldsymbol{u}^{t+1}$ and $\boldsymbol{v}^t$ converges weakly to their Gaussian version: $\mathbf{U}^t = m_u^t \mathbf{u}^* + \sqrt{q_u^t} \boldsymbol{z}$ (with $\boldsymbol{z}$ random Gaussian as above) and $\mathbf{V}^t = \sqrt{q_v^t - m_v^t} \mathbf{w} + m_v^t A^T \mathbf{u}^*$ (with $\mathbf{w} \sim \mathcal{N}(0, \mathbb{I}_N)$), in the sense that, for any deterministic sequence $\phi_n : (\mathbb{R}^D \times \mathbb{R}^N)^t \times \mathbb{R}^N \to \mathbb{R}$ of uniformly pseudo-Lipschitz functions of order $k$,*

$$
\phi_n(\mathbf{u^0}, \mathbf{v^0}, \mathbf{u^1}, \mathbf{v^1}, \ldots, \mathbf{v^{t-1}}, \mathbf{u^t}) \stackrel{\mathbb{P}}{\simeq} \mathbb{E}_{\mathbf{U}, \mathbf{V}} \left[ \phi_\mathbf{n} \left( \mathbf{U^0}, \mathbf{V^0}, \mathbf{U^1}, \mathbf{V^1}, \ldots, \mathbf{V^{t-1}}, \mathbf{U^t} \right) \right].
\tag{33}
$$

An immediate corollary we shall use is the following:

**Corollary 2 (Fixed point initialization)** *Consider a fixed point of the state evolution equations (31) (we denote the fixed point quantities by $\mathrm{fp}$). If one initializes an AMP sequence in the fixed point, i.e. by using $\mathbf{u^0} = m_u^{\mathrm{fp}} \mathbf{u}^* + \sqrt{q_u^{\mathrm{fp}}} \boldsymbol{z}$ (with $\boldsymbol{z}$ random Gaussian as above), then in probability for any time $t > 0$ and as $d \to \infty$, the sequence of iterates $\boldsymbol{U}^t$ and $\boldsymbol{V}^t$ remains in their fixed point.*

We then use the following theorem, that ensure with the initialization of 2, then the GAMP equation converges to their fixed point:

**Theorem 5 (Convergence of GAMP, Lemma 7 from [Loureiro et al., 2021b])** *Consider the GAMP iteration with $e$, $g$ as in (28), where all free parameters are initialized at any fixed point of the state evolution equations. If the associated loss (27) is strongly convex, then GAMP converges to the fixed point of the loss under study:*

$$
\lim_{t \to \infty} \lim_{d \to \infty} \frac{1}{\sqrt{d}} \|\boldsymbol{u}^t - \boldsymbol{u}^{\mathrm{fp}}\|_F = 0, \quad \lim_{t \to \infty} \lim_{d \to \infty} \frac{1}{\sqrt{d}} \|\boldsymbol{v}^t - \boldsymbol{v}^{\mathrm{fp}}\|_F = 0
\tag{34}
$$

Thus, state evolution allows to compute all scalar observables (such as overlaps, errors, etc) on the iterates, which at convergence and under the setting of Appendix A.3.1, are equivalent to the global minimum of (27).

## A.4 State evolution for Theorem 1

Now that we have reviewed how to use GAMP to study the fixed point of an optimization problem, it remains to show that the present matrix problem can be mapped to such an equivalent vectorized problem. Consider (9), and write it in the following form:

$$\hat{S} = \arg\min_{S \in \mathcal{C}} \tilde{\mathcal{L}}(S), \ \ \text{with} \ \ \tilde{\mathcal{L}}(S) := \sum_{\mu=1}^{n} \ell\left(\mathrm{Tr}[X^\mu S], \mathrm{Tr}[X^\mu S^*]\right) + R(S), \tag{35}$$

where $S \in \mathrm{Sym}_d$ is a symmetric $d \times d$ matrix, $\mathcal{C} \subseteq \mathbb{R}^D$ is convex and $\ell, R$ are convex in their first argument.

### A.4.1 Mapping matrix-GLM to vector-GLM

We consider the mapping from vec : $\mathrm{Sym}_d \to \mathbb{R}^{d(d+1)/2}$ (which conveniently maps the Frobenius scalar product in $\mathrm{Sym}_d$ given by $\langle A\,; B\rangle = \mathrm{Tr}(AB)$ to the standard Euclidean scalar product in $\mathbb{R}^{d(d+1)/2}$) given by

$$\mathrm{vec}\,(A)_{(ab)} = \left\langle b^{(ab)}\,; A\right\rangle = \sqrt{2 - \delta_{ab}} A_{ab}\,, \tag{36}$$

under the choice of orthonormal basis

$$b^{(aa)}_{ij} = \delta_{ia}\delta_{ja}\,, \quad b^{(ab)}_{ij} = \frac{\delta_{ia}\delta_{jb} + \delta_{ib}\delta_{ja}}{\sqrt{2}}\,. \tag{37}$$

Here $(ab)$ stands for the ordered pair of $1 \le a \le b \le d$, and we denote $A_{ij}$ as the $i, j$ entry of a matrix $A$, while as $A_{(ab)}$ the component of matrix $A$ onto the basis element $b^{(ab)}$. Let us denote $d(d+1)/2 = D$ (we will often use $D \approx d^2/2$ as we will be interested in the leading order in $d$).

Under this mapping, we have that $\mathbf{w} = \sqrt{d/2}\mathrm{vec}\,(S)$ satisfies

$$||\mathbf{w}||^2 = \frac{d}{2}\,\mathrm{Tr}[S^2] = \mathcal{O}(D)\,, \tag{38}$$

for any $S$ with asymptotically well-defined spectral density. In particular in state evolution $\boldsymbol{u}^* = \mathrm{vec}\,(S^*)$ is such that $||\boldsymbol{u}^*||^2/D = Q^*$. Moreover, we can define the correctly normalized sensing vectors

$$A^\mu_{(ab)} := \sqrt{\frac{1}{d+1}}\mathrm{vec}\,(X^\mu)_{(ab)} = \sqrt{\frac{d(2 - \delta_{ab})}{2D}}X^\mu_{ab} \sim \mathcal{N}(0, 1/D)\,, \tag{39}$$

for all $1 \le a \le b \le d$, where we used that $X \sim \mathrm{GOE}(d)$, giving

$$\mathrm{Tr}[X^\mu S] = \sum_{(ab)} \mathrm{vec}\,(X^\mu)_{(ab)}\mathrm{vec}\,(S)_{(ab)} = \sqrt{2\frac{d+1}{d}}\sum_{(ab)} A^\mu_{(ab)}\mathbf{w}_{(ab)} \approx \sqrt{2}\sum_{(ab)} A^\mu_{(ab)}\mathbf{w}_{(ab)}\,. \tag{40}$$

where $\alpha = n/d^2$.

Thus, we can pick

$$\tilde{\ell}(a, b) = \ell(\sqrt{2}a, \sqrt{2}b)\,, \qquad \tilde{R}(\mathbf{w}) = R(\mathrm{mat}\,(\mathbf{w})/\sqrt{d/2}) \quad \text{and} \quad \phi(z) = z + \sqrt{\Delta/2}\,\xi \tag{41}$$

where mat () denotes the inverse of vec (), $\xi$ is a standard Gaussian (we are restricting here to the Gaussian noise function $\phi$), and similarly we define $\tilde{\mathcal{C}}$ by a rescaling of $\mathcal{C}$. Finally, we define the equivalent vector problem in dimension $D = d(d+1)/2$ with $N = n$ samples

$$\hat{\mathbf{w}} = \arg\min_{\mathbf{w} \in \tilde{\mathcal{C}}} \tilde{\mathcal{L}}_{\mathrm{vec}}(\mathbf{w}), \ \ \text{with} \ \ \tilde{\mathcal{L}}_{\mathrm{vec}}(\mathbf{w}) := \sum_{\mu=1}^{N} \tilde{\ell}\left(\boldsymbol{a}^\mu \cdot \mathbf{w}, \phi(\boldsymbol{a}^\mu \cdot \mathbf{w}^*)\right) + \tilde{R}(\mathbf{w})\,. \tag{42}$$

In particular, the GAMP iteration described in Appendix A.3 for this loss solves (9) modulo the bijection vec (), with generic convex loss, regularization and constraint set $\mathcal{C}$.

### A.4.2 GAMP denoiser $g$ for the square loss

We directly specialize to the case of the square loss, but all this can be generalized to generic losses, see [Loureiro et al., 2021a]. For $\ell(a, b) = (a - b)^2$ we have

$$\bar{g}(r, s, \bar{b}) = \underset{z \in \mathbb{R}}{\arg\min} \left[ \tilde{\ell}(z, s) + \frac{1}{2\bar{b}}(z - r)^2 \right] = \frac{r + 4\bar{b}s}{1 + 4\bar{b}} \quad \text{and} \quad g(r, s, \bar{b}) = \frac{s - r}{\bar{b} + 1/4}. \tag{43}$$

The associated state evolution equations read (recall that $\phi(z^*) = z^* + \sqrt{\Delta/2}\xi$, including the average over the noise $\xi$ in the activation $\phi$ in $\mathbb{E}_u$)

$$\begin{cases} d^{t+1} &= -\frac{N}{D}\mathbb{E}_u^t \partial_\omega g(\omega, \phi(z^*), b^t) = 2\alpha \frac{1}{b^t + 1/4} \\ m_u^{t+1} &= \frac{N}{D}\mathbb{E}_u^t [\partial_{z^*} g(\omega, \phi(z^*), b^t)] = 2\alpha \frac{1}{b^t + 1/4} \\ q_u^{t+1} &= \frac{N}{D}\mathbb{E}_u^t [(g(\omega, \phi(z^*), b^t)^2] = 2\alpha \frac{Q^* - 2m + q + \Delta/2}{(b^t + 1/4)^2} \end{cases} \tag{44}$$

where we used that $N/D = 2n/d^2 = 2\alpha$.

### A.4.3 GAMP denoiser $e$ for spectral regularization, including PSD constraints and nuclear/Frobenius regularization

**Spectral denoising function $e$.** We have (calling $\Gamma = \text{mat}\left(\sqrt{2/d}r\right)$ and $k = \bar{d}$ to avoid confusion with the dimension $d$)

$$\begin{aligned} e(\boldsymbol{r}, k) &= \underset{\boldsymbol{z} \in \tilde{\mathcal{C}}}{\arg\min} \left[ \tilde{R}(\boldsymbol{z}) + \frac{k}{2}||\boldsymbol{z} - k^{-1}\boldsymbol{r}||^2 \right] \\ &= \underset{\boldsymbol{z} \in \tilde{\mathcal{C}}}{\arg\min} \left[ R(\sqrt{2/d}\text{mat}\,(\boldsymbol{z})) + \frac{k}{2}||\boldsymbol{z} - k^{-1}\boldsymbol{r}||^2 \right] \\ &= \sqrt{\frac{d}{2}}\text{vec}\left( \underset{Z \in \mathcal{C}}{\arg\min} \left[ R(Z) + \frac{d^2 k}{4}||Z - k^{-1}\Gamma||_F^2 \right] \right) \end{aligned} \tag{45}$$

Now, assume that both $\mathcal{C}$ and $R$ are spectral, meaning that they do not depend on the eigenvectors of $Z$. Then, if we consider the spectral decompositions $\Gamma = ODO^T$, $e(\boldsymbol{r}, k) = ULU^T$, and we parametrize $Z = VTV^T$, then for each $T$

$$U = \underset{V \in O(d)}{\arg\max} \text{Tr}(VLV^T ODO^T), \tag{46}$$

which, by Von Neumann's trace inequality [Mirsky, 1975] is given by the rotation $V$ such that $U^T O$ aligns the eigenvalues of $T$ and $D$ in decreasing order, which if we take the convention that all spectral decomposition are given with eigenvalues sorted decreasingly, gives $U = O$. Thus, the problem reduces to a spectral minimization

$$L = \underset{T \in \mathcal{C}}{\arg\min} \left[ d^{-2}R(T) + \frac{k}{4}\sum_{i=1}^d T_i^2 - \frac{1}{2}\sum_{i=1}^d T_i D_i \right], \tag{47}$$

where here we are abusing the notation by writing $T_i$ to mean $T_{ii}$ as $T$ is really a diagonal $d \times d$ matrix, and similarly for $D$. Notice that this expression is still general.

**Spectral denoising function $e$ for the case of the main text.** We now specialize to the case of the main text. If $\mathcal{C}$ is the set of PSD matrices and $R(T) = d^2(\tilde{\lambda}\sum_i T_i + \tau/2\sum_i T_i^2)$, then

$$L_i = \frac{1}{2\tau + k}\text{ReLU}\left(D_i - 2\tilde{\lambda}\right). \tag{48}$$

In the end, this means that if

$$\boldsymbol{r} = \text{vec}\left(\sqrt{d/2}\,ODO^T\right) \tag{49}$$

with eigenvalues $D_i$ sorted decreasingly, then

$$e(\boldsymbol{r}, k) = \sqrt{\frac{d}{2}}\text{vec}\left(O\frac{1}{2\tau + k}\text{ReLU}\left(D - 2\tilde{\lambda}\right)O^T\right) \tag{50}$$

where here ReLU is applied component-wise to the diagonal matrix $D - 2\tilde{\lambda}\mathbb{I}_d$.

**State evolution equations.** Now we need to derive explicit expressions for the associated state evolution equations

$$
\begin{cases}
m_v^t & = \frac{1}{D}\mathbb{E}_v^t\left[\left\langle \mathbf{x_0}\,;e(\sqrt{q_u^t}\mathbf{z}+m_u^t\mathbf{u}^*,d^t)\right\rangle\right] \\
q_v^t & = \frac{1}{D}\mathbb{E}_v^t\left[\left\langle e(\sqrt{q_u^t}\mathbf{z}+m_u^t\mathbf{u}^*,d^t)\,;e(\sqrt{q_u^t}\mathbf{z}+m_u^t\mathbf{u}^*,d^t)\right\rangle\right] \\
b^t & = \frac{1}{D}\mathbb{E}_v^t\sum_{i=1}^D \partial_i e(\sqrt{q_u^t}\mathbf{z}+m_u^t\mathbf{u}^*,d^t)_i
\end{cases}
\tag{51}
$$

We revert to the case of general $\mathcal{C}$ and $R$ for this, assuming that $R$ is strictly convex over $\mathcal{C}$ for simplicity. It is useful to consider the following function

$$
\begin{aligned}
\Psi(k,q_u,m_u) &= \frac{1}{D}\mathbb{E}_v^t \min_{\mathbf{w}\in\mathcal{C}}\left[\tilde{R}(\mathbf{w})+\frac{k}{2}||\mathbf{w}||^2-\sqrt{q_u}\langle \mathbf{w}\,;\mathbf{z}\rangle-m_u\langle \mathbf{w}\,;\mathbf{u}^*\rangle\right] \\
&= -\frac{1}{D}\lim_{\beta\to+\infty}\frac{1}{\beta}\mathbb{E}_v^t \log\int_{\mathbf{w}\in\tilde{\mathcal{C}}} d\mathbf{w}\, e^{-\beta\left(\tilde{R}(\mathbf{w})+\frac{k}{2}||\mathbf{w}||^2-\sqrt{q_u}\langle \mathbf{w}\,;\mathbf{z}\rangle-m_u\langle \mathbf{w}\,;\mathbf{u}^*\rangle\right)} \\
&= -\frac{1}{D}\lim_{\beta\to+\infty}\mathbb{E}_v^t\frac{1}{\beta}\log \mathcal{Z}_\beta(k,q_u,m_u,\mathbf{z},\mathbf{x}_0)\,,
\end{aligned}
\tag{52}
$$

where the limiting procedure and all following integration/derivation/limit exchanges are well defined by the strict convexity of $R$, and $\mathcal{Z}_\beta$ was defined in the last line. Then we have immediately

$$
\begin{aligned}
-\partial_{m_u}\Psi(k,q_u,m_u) &= \frac{1}{D}\mathbb{E}_v^t\left[\left\langle \mathbf{u}^*\,;e(\sqrt{q_u^t}\mathbf{z}+m_u^t\mathbf{u}^*,d^t)\right\rangle\right]\,, \\
2\partial_k\Psi(k,q_u,m_u) &= \frac{1}{D}\mathbb{E}_v^t\left[\left\langle e(\sqrt{q_u^t}\mathbf{z}+m_u^t\mathbf{u}^*,d^t)\,;e(\sqrt{q_u^t}\mathbf{z}+m_u^t\mathbf{u}^*,d^t)\right\rangle\right]\,.
\end{aligned}
\tag{53}
$$

Moreover (dropping the omnipresent $\beta$ limit and average $\mathbb{E}$, as well as the inputs of $\mathcal{Z}$ for readability)

$$
\begin{aligned}
&-2\partial_{q_u}\Psi(k,q_u,m_u) = \\
&= \frac{1}{D}\mathbb{E}_v^t\frac{1}{\sqrt{q_u}}\int_{\mathbf{w}\in\tilde{\mathcal{C}}} d\mathbf{w}\sum_i w_i z_i \frac{e^{-\beta\left(\tilde{R}(\mathbf{w})+\frac{k}{2}||\mathbf{w}||^2-\sqrt{q_u}\langle \mathbf{w}\,;\mathbf{z}\rangle-m_u\langle \mathbf{w}\,;\mathbf{u}^*\rangle\right)}}{\mathcal{Z}_\beta} \\
&= \frac{1}{D}\mathbb{E}_v^t\frac{1}{\sqrt{q_u}}\int_{\mathbf{w}\in\tilde{\mathcal{C}}} d\mathbf{w}\sum_i w_i\partial_{z_i}\frac{e^{-\beta\left(\tilde{R}(\mathbf{w})+\frac{k}{2}||\mathbf{w}||^2-\sqrt{q_u}\langle \mathbf{w}\,;\mathbf{z}\rangle-m_u\langle \mathbf{w}\,;\mathbf{u}^*\rangle\right)}}{\mathcal{Z}_\beta} \\
&= \frac{1}{D}\mathbb{E}_v^t\sum_i \partial_{\sqrt{q_u}z_i+m_u u_i^*}\int_{\mathbf{w}\in\tilde{\mathcal{C}}} d\mathbf{w}\, w_i\frac{e^{-\beta\left(\tilde{R}(\mathbf{w})+\frac{k}{2}||\mathbf{w}||^2-\sqrt{q_u}\langle \mathbf{w}\,;\mathbf{z}\rangle-m_u\langle \mathbf{w}\,;\mathbf{u}^*\rangle\right)}}{\mathcal{Z}_\beta} \\
&= \frac{1}{D}\mathbb{E}_v^t\sum_i \partial_{r_i}e(\mathbf{r},k)_i|_{\mathbf{r}=\sqrt{q_u}z_i+m_u u_i^*} \\
&= \frac{1}{D}\mathbb{E}_v^t\sum_{i=1}^D \partial_i e(\sqrt{q_u^t}\mathbf{z}+m_u^t\mathbf{u}^*,d^t)_i
\end{aligned}
\tag{54}
$$

where in the third line we used Stein's lemma, so that

$$
\begin{cases}
m_v^t & = -\partial_{m_u}\Psi(d^t,q_u^t,m_u^t) \\
q_v^t & = 2\partial_k\Psi(d^t,q_u^t,m_u^t) \\
b^t & = -2\partial_{q_u}\Psi(d^t,q_u^t,m_u^t)
\end{cases}\,.
\tag{55}
$$

We just need to compute $\Psi$ for the sake of the state evolutions.

**State evolution equations for the case of the main text.** If $\mathcal{C}$ is the set of PSD matrices and $R(T)=d^2(\tilde{\lambda}\sum_i T_i+\tau/2\sum_i T_i^2)$, then

$$
\Psi(k,q_u,m_u) = 2\mathbb{E}_v^t\left[\sum_i L_i+\tau/2\sum_i L_i^2+\frac{k}{4}\sum_i L_i^2-\frac{m_u}{2}\sum_i L_i D_i\right]\,,
\tag{56}
$$

where $D_i$ is the $i$-th eigenvalue of the matrix $S^*+\sqrt{q_u}/m_u Z$, and $L_i$ is determined by

$$
L_i = \frac{m_u}{2\tau+k}\mathrm{ReLU}\left(D_i-2\tilde{\lambda}/m_u\right)\,,
\tag{57}
$$

giving

$$\Psi(k, q_u, m_u) = 2\mathbb{E}_v^t\left[\tilde{\lambda}\sum_i L_i + \tau/2\sum_i L_i^2 + \frac{k}{4}\sum_i L_i^2 - \frac{m_u}{2}\sum_i L_i D_i\right]$$

$$= -\frac{m_u^2}{2(k+2\tau)}\mathbb{E}_v^t\sum_i \text{ReLU}(D_i - 2\tilde{\lambda}/m_u)^2$$

$$\approx -\frac{m_u^2}{2(k+2\tau)}\int_{2\tilde{\lambda}/m_u} dx\,\mu^*_{\sqrt{q_u}/m_u}(x)(x - 2\tilde{\lambda}/m_u)^2 \qquad (58)$$

$$= -\frac{m_u^2}{2(k+2\tau)}J(\sqrt{q_u}/m_u, 2\tilde{\lambda}/m_u)\,,$$

where in the last step we used that at leading order the spectrum of $S^* + \sqrt{q_u}/m_u Z$ concentrates (by assumption in 1) to $\mu^*_{\sqrt{q_u}/m_u}$ defined in Theorem 1.

### A.4.4 Final form of the state evolution and observables

Collecting the state evolution equations derived in Appendices A.4.2 and A.4.3, calling $m_v = m$, $q_v = q$, $b = \Sigma$, $m_u = \hat{m}$, $q_u = \hat{q}$ and $d = \hat{\Sigma}$ we get the system of equations

$$\begin{cases} \hat{\Sigma}^{t+1} &= \frac{2\alpha}{\Sigma^t+1/4} \\ \hat{m}^{t+1} &= \frac{2\alpha}{\Sigma^t+1/4} \\ \hat{q}^{t+1} &= 2\alpha\frac{Q^*-2m+q+\Delta/2}{(\Sigma^t+1/4)^2} \\ m^t &= \partial_{\hat{m}^t}\frac{(\hat{m}^t)^2}{2(\hat{\Sigma}^t+2\tau)}J(\sqrt{\hat{q}^t}/\hat{m}^t, 2\tilde{\lambda}/\hat{m}^t) \\ q^t &= \frac{(\hat{m}^t)^2}{(\hat{\Sigma}^t+2\tau)^2}J(\sqrt{\hat{q}^t}/\hat{m}^t, 2\tilde{\lambda}/\hat{m}^t) \\ \Sigma^t &= \frac{(\hat{m}^t)^2}{\hat{\Sigma}^t+2\tau}\partial_{\hat{q}^t}J(\sqrt{\hat{q}^t}/\hat{m}^t, 2\tilde{\lambda}/\hat{m}^t) \end{cases} \qquad (59)$$

where $J$ is defined in Theorem 1. This reduces to the equations presented in in Theorem 1 at its fixed point and under the change of variable $\delta = \sqrt{\hat{q}}/\hat{m}, \epsilon = 2/\hat{m}$.

The test error then is given by

$$e_{\text{test}} = Q_0 - 2m + q = 2\alpha\delta^2 - \Delta/2\,. \qquad (60)$$

The expression train loss instead van be derived by using the fact that the residuals in the data part of the loss can be computed from state evolution as $\hat{q}/16$, and by evaluating the regularization at the spectral density $\mu^*_\delta$ (which is just the Tr of a shift and rescaling of this spectral distribution) giving the expression in Theorem 1 in the limit $\tau \to 0^+$.

Notice that these state evolution can be also heuristically applied in the case $\tilde{\lambda} = 0$ as long as the problem has a unique global minimum. This can be checked by having $\Sigma < +\infty$, as $\Sigma$ is a proxy for the inverse of the curvature of the loss at the global minimum [Clarté et al., 2023], which becomes flat when $\Sigma = +\infty$. Notice also that $\Sigma$ can be used to guarantee, at the heurisitc level, that the minimum of the global loss is unique even without the need of the regularization $\tau > 0$.

### A.5 Replicon condition for Generalized Approximate Message Passing fixed points

For completeness, we discuss in this section the pointwise convergence of the GAMP algorithm when initialized at a fixed point of the state evolution equations. While the convex nature of the problem ensure this convergence [Loureiro et al., 2021b], we discussed here the explicit criterion that can also be directly checked within the state evolution formalism. As noted by Bolthausen [2014], it hinges on a stability condition—known in statistical physics as the *replicon* or de Almeida-Thouless (AT) condition [de Almeida and Thouless, 1978, De Dominicis and Kondor, 1983]—which can be directly and easily checked from the state evolution equations.

We thus provide the replicon condition on point-wise convergence of GAMP for generic non-separable denoisers (see for instance Vilucchio et al. [2025] for the separable case).

Consider again the recursion (25)

$$
\begin{aligned}
\boldsymbol{u}^{t+1} &= A^T g_t(\boldsymbol{v}^t) + d_t e_t(\boldsymbol{u}^t), \\
\boldsymbol{v}^t &= A e_t(\boldsymbol{u}^t) - b_t g_{t-1}(\boldsymbol{v}^{t-1}),
\end{aligned}
\tag{61}
$$

with the $b_t, d_t$ given in (26). We want to assess linear stability. To this end, we suppose that the iteration is initialized at a fixed point $(\boldsymbol{u}, \boldsymbol{v})$ (and that at this point we also assume that the nonlinearities $g, e$ and the Onsager's terms are constant in time), consider a Gaussian perturbation $\boldsymbol{\epsilon} \sim \mathcal{N}(0, \epsilon \mathbb{I}_d)$ of $\boldsymbol{u}$, and compute whether the iteration converges back to the fixed point after such perturbation. Notice that we can avoid perturbing the Onsager's terms, as in the high-dimensional limit they are independent on the iterates. We have (at first order in $\epsilon$)

$$
\begin{aligned}
\boldsymbol{v}^{\mathrm{new}} &= A e(\boldsymbol{u} + \boldsymbol{\epsilon}) - b g(\boldsymbol{v}) = \boldsymbol{v} + A(\nabla e(\boldsymbol{u}))\boldsymbol{\epsilon}, \\
\boldsymbol{u}^{\mathrm{new}} &= A^T g(\boldsymbol{v} + A(\nabla e(\boldsymbol{u}))\boldsymbol{\epsilon}) + d\, e(\boldsymbol{u} + \boldsymbol{\epsilon}) \\
&= \boldsymbol{u} + A^T \nabla(g(\boldsymbol{v}))A(\nabla e(\boldsymbol{u}))\boldsymbol{\epsilon} - D^{-1}(\nabla \cdot g(\boldsymbol{v}))(\nabla e(\boldsymbol{u}))\boldsymbol{\epsilon}, \\
u_k^{\mathrm{new}} &= u_k + \sum_{i,j=1}^{D} \sum_{\mu,\nu=1}^{N} A_{\nu k} A_{\mu j} \partial_\mu g_\nu \partial_i e_j \epsilon_i - \frac{1}{N} \sum_{\mu=1}^{N} \partial_\mu g_\mu \sum_{i=1}^{D} \partial_i e_k \epsilon_i \\
&= u_k + \sum_{i,j=1}^{D} \sum_{\mu,\nu=1}^{N} (A_{\nu k} A_{\mu j} - D^{-1}\delta_{\mu\nu}\delta_{kj})\partial_\mu g_\nu \partial_i e_j \epsilon_i,
\end{aligned}
\tag{62}
$$

where in the second to last step we used the explicit form of the Onsager's term $d$, and in the last step we suppressed the dependencies of the functions $g, e$ on $\boldsymbol{u}, \boldsymbol{v}$ for clarity and passed in components notation.

The L2 norm of the perturbation $\boldsymbol{u}^{\mathrm{new}} - \boldsymbol{u}$ equals (on average over the initial perturbation $\boldsymbol{\epsilon}$)

$$
\begin{aligned}
\mathbb{E}_{\boldsymbol{\epsilon}}(u_k^{\mathrm{new}} - u_k)^2 &= \mathbb{E}_{\boldsymbol{\epsilon}} \sum_{i,j=1}^{D} \sum_{\mu,\nu=1}^{N} (A_{\nu k} A_{\mu j} - N^{-1}\delta_{\mu\nu}\delta_{kj})\partial_\mu g_\nu \partial_i e_j \epsilon_i \\
&\qquad \times \sum_{i',j'=1}^{D} \sum_{\mu',\nu'=1}^{N} (A_{\nu' k} A_{\mu' j'} - N^{-1}\delta_{\mu'\nu'}\delta_{kj'})\partial_{\mu'} g_{\nu'} \partial_{i'} e_{j'} \epsilon_{i'} \\
&= N^{-2} \sum_{i,j,j'=1}^{D} \sum_{\mu,\nu,\mu',\nu'=1}^{N} \delta_{\nu\nu'}\delta_{jj'}\delta_{\mu\mu'}\partial_\mu g_\nu \partial_i e_j \partial_{\mu'} g_{\nu'} \partial_{i'} e_{j'} \\
&\approx \frac{1}{D} \sum_{i,j=1}^{D} (\partial_i e_j)^2 \times \frac{1}{D} \sum_{\mu,\nu=1}^{N} (\partial_\mu g_\nu)^2,
\end{aligned}
\tag{63}
$$

where we used that $\mathbb{E}_{\boldsymbol{\epsilon}} \epsilon_i \epsilon_{i'} = \delta_{ii'}$, and the law of large number to substitute

$$
\begin{aligned}
&(A_{\nu k} A_{\mu j} - D^{-1}\delta_{\mu\nu}\delta_{kj})(A_{\nu' k} A_{\mu' j'} - N^{-1}\delta_{\mu'\nu'}\delta_{kj'}) \\
&\approx D^{-2}\delta_{\nu\nu'}\delta_{jj'}\delta_{\mu\mu'},
\end{aligned}
\tag{64}
$$

where in the last step we only kept the term that will contribute to leading order (equality at eluding order is denoted by $\approx$). This is the generalization to non-separable functions $g, e$ of [Vilucchio et al., 2025, Definition 1].

Thus, the linear stability criterion for GAMP is

$$
\frac{N}{D} \times \frac{1}{D} \sum_{i,j=1}^{D} (\partial_i e_j)^2 \times \frac{1}{N} \sum_{\mu,\nu=1}^{N} (\partial_\mu g_\nu)^2 < 1.
\tag{65}
$$

In particular, in the case of independent observations we will have $\partial_\mu g_\nu = \delta_{\mu\nu}\partial_\mu g_\mu$, and in the case of separable denoiser one would have $\partial_i e_j = \delta_{ij}\partial_i e_i$. Notice also that this criterion is independent of the value of the iterates $\boldsymbol{u}, \boldsymbol{v}$ at the fixed points, as it concentrates onto its state-evolution-predicted value.

In the case of $\ell_2$ loss +PSD + nuclear norm regularization, a fixed point of the state evolution describes a stable fixed point of GAMP if

$$\int dx\,\mu_\delta^*(x)\int dy\,\mu_\delta^*(y)\left(\frac{\mathrm{ReLU}(x-\epsilon)-\mathrm{ReLU}(y-\epsilon)}{x-y}\right)^2<2\alpha\,. \tag{66}$$

We have checked numerically that this condition was satisfied in our fixed point (in accord with Theorem 5).

## B Minimal norm interpolation limit $\lambda \to 0^+$

Consider (6). We would like to give a prescription to manipulate this equations in order to describe minimal regularization interpolators. We first remark that the limit $\lambda \to 0^+$ of (6) as written describes the case of non-regularized minimization, and thus Theorem 1 will be valid in this case only for $\alpha > \alpha_{\mathrm{interp}}$, when only one interpolator of the training dataset exists.

Consider now the rescaling of (9) given by $\tilde{\lambda}^{-1}\tilde{\mathcal{L}}$. For any fixed positive $\tilde{\lambda}$, this rescaling does not alter the solution of the minimization problem. In the limit $\tilde{\lambda} \to 0^+$ instead, the loss first imposes that the global minimum is an interpolator of the training dataset, and then minimizes the regularization within the interpolator set. Thus, rescaling the loss by $\tilde{\lambda}^{-1}$ and *then* sending $\tilde{\lambda} \to 0^+$ will describe minimal regularization interpolators, as long as $\alpha < \alpha_{\mathrm{interp}}$ in the noisy case (after that threshold no interpolator exists, and the minimization problem is ill-defined) of for all $\alpha$ in the noiseless case (as in this case there exists always one interpolator).

We practically achieve the loss rescaling in (6) by renaming $\tilde{\lambda}\epsilon \to \epsilon$, and obtaining the new system of equations

$$\begin{cases} 4\alpha\delta - \tilde{\lambda}\frac{\delta}{\epsilon} = \partial_1 J(\delta,\epsilon) \\ Q^* + \frac{\Delta}{2} + 2\alpha\delta^2 - \tilde{\lambda}\frac{\delta^2}{\epsilon} = (1-\epsilon\partial_2)J(\delta,\epsilon) \end{cases}, \tag{67}$$

equivalent to (6) for all $\tilde{\lambda} > 0$, and then taking $\tilde{\lambda} \to 0^+$.

Thus, to summarize the learning curves at $\tilde{\lambda} = 0^+$ are found as follows.

- In the noiseless case $\Delta = 0$, we solve (67) with $\tilde{\lambda} = 0$ for all values of $\alpha$.

- In the noisy case $\Delta > 0$, we solve (67) with $\tilde{\lambda} = 0$ for all values of $\alpha < \alpha_{\mathrm{inter}}$, and we solve (6) for $\tilde{\lambda} = 0$ for all values of $\alpha > \alpha_{\mathrm{inter}}$.

## C Derivation of Result 1 and Corollary 1

### C.1 Prerequisites

For both Result 1 and Corollary 1 we will need the following results on the spectral density $\mu^* \boxplus \mu_{\mathrm{s.c.},\delta}$, where $\boxplus$ is the free convolution and $\mu_{\mathrm{s.c.},\delta} = \sqrt{4-x^2}/(2\pi\delta^2)$ the semicircle distribution of radius $2\delta$ for $\delta > 0$. Here $\mu^*(x) = \sqrt{\kappa^*}\mu_{\mathrm{M.P.}}(\sqrt{\kappa^*}x)$, where $\mu_{\mathrm{M.P.}}$ is the Marchenko-Pastur distribution [Marchenko and Pastur, 1967] with parameter $\kappa^*$ (the asymptotic spectral distribution of $A^T A/m$ where $A \in \mathbb{R}^{m\times d}$ has i.i.d. zero-mean unit-variance components), and $Q^* = 1 + \kappa^*$.

We start by recalling that

$$J(\delta,s) = \int_s^{+\infty} dx\,\mu_\delta^*(x)\,(x-s)^2 = Q^* + \delta^2 - 2sM^* + s^2 - \int_{-\infty}^s dx\,\mu_\delta^*(x)\,(x-s)^2, \tag{68}$$

where

$$Q^* = \int dx\,\mu^*(x)\,x^2\,, \quad M^* = \int dx\,\mu^*(x)\,x\,, \tag{69}$$

and where we used that

$$\int dx\,\mu_\delta^*(x)\,x = \int dx\,\mu^*(x)\,x + \int dx\,\mu_{\mathrm{s.c.},\delta}\,x = M^*$$

$$\int dx\,\mu_\delta^*(x)\,x^2 = \int dx\,\mu^*(x)\,x^2 + \int dx\,\mu_{\mathrm{s.c.},\delta}\,x^2 = Q^* + \delta^2 \tag{70}$$

by additivity of the mean and variance.

Now we use [Maillard et al., 2024, Appendix D.3] to state that for any fixed $x$, for $\delta \to 0^+$, we have at leading order
$$\delta\sqrt{1 - \kappa^*}\,\mu_\delta^*(x\delta\sqrt{1 - \kappa^*}) \approx (1 - \kappa^*)\mu_{\text{s.c.}}(x)\,. \tag{71}$$
This will allow us to compute the last integral in (68) when $\delta \ll 1$ for some values of $s$. Indeed, when $\delta \ll 1$ and $0 < \kappa^* < 1$ the spectral density $\mu_\delta^*$ is composed by a small semicircle around the origin with mass $1 - \kappa^*$, and an approximately $\mu^*$-shaped bulk gapped away from the origin. So as long as $s$ lies in the gap between the two bulks, only this semicircle will contribute to (68). For $\kappa^* > 1$ instead, the bulk at the origin is not present leading to a zero contribution from all the integrals. The case $\kappa^* = 1$ is more delicate, as there the spectral distribution of the target may be non-gapped (for example, in the narrow target case). We recover this case by a limiting procedure from the two sides.

## C.2 Derivation of Result 1

To find the interpolation threshold, we consider (6) for $\tilde\lambda = 0$. To do this, we need to make sure that we are in the $\alpha > \alpha_{\text{inter}}$ region, where only one interpolator (noiseless case) or no interpolator (noisy case) exists, giving a single minimum of the training loss. This can be checked by having
$$\Sigma = \alpha\epsilon - \frac{1}{4} < +\infty\,, \tag{72}$$
where $\Sigma$ was defined in (59). The interpolation threshold will be exactly at the point where $\Sigma \propto \epsilon$ diverges.

**Noisy case.** Consider the (6) with $\tilde\lambda = 0$. We have
$$\begin{cases} 4\alpha\delta - \frac{\delta}{\epsilon} = \partial_1 J(\delta, 0) \\ Q^* + \frac{\Delta}{2} + 2\alpha\delta^2 - \frac{\delta^2}{\epsilon} = J(\delta, 0) \end{cases}. \tag{73}$$

We now consider the limit $\epsilon \to +\infty$, obtaining
$$\begin{cases} 4\alpha\delta = \partial_1 J(\delta, 0) \\ Q^* + \frac{\Delta}{2} + 2\alpha\delta^2 = J(\delta, 0) \end{cases} \implies \begin{cases} \alpha_{\text{inter}} = \frac{1}{4\delta}\partial_1 J(\delta, 0) \\ Q^* + \frac{\Delta}{2} = J(\delta, 0) - \frac{\delta}{2}\partial_1 J(\delta, 0) \end{cases}, \tag{74}$$

i.e. an equation for $\delta$, which can be plugged in to directly find $\alpha$. This gives the first part of Result 1. Notice that the value of $\delta$ found will be finite, giving also an analytic prediction of the height of the cusp of the test error at interpolation.

For $\Delta \to +\infty$, we see that a consistent solution is given by $\delta \to +\infty$ (starting from (73) at finite $\epsilon$, and then taking the limit). In that case, $\mu_\delta^*$ is, at leading order, a semicircle with radius $2\delta$, so that
$$J(\delta, 0) \approx \frac{1}{2}\delta^2 \quad \text{and} \quad \partial_1 J(\delta, 0) \approx \delta\,, \tag{75}$$

implying that for large $\Delta$ the interpolation threshold converges to $1/4$.

**Noiseless case.** Here we can advance more. In the noiseless case $\Delta \to 0$, we also know that after interpolation we have zero test error, hence $\delta = 0$. Thus, we can expand $J(\delta, 0)$ for small $\delta$. Using (68) and (71) we have
$$\begin{aligned} \partial_\delta J(\delta, 0) &= \partial_\delta(Q^* + \delta^2) - \partial_\delta \int_{-\infty}^0 dx\,\mu_\delta^*(x)\,x^2 \\ &= 2\delta - \partial_\delta\left[\delta^2(1 - \kappa^*)\int_{-\infty}^0 dy\,\delta\sqrt{1 - \kappa^*}\mu_\delta^*(\delta\sqrt{1 - \kappa^*}y)\,y^2\right] \\ &\approx 2\delta - \partial_\delta\left[\delta^2(1 - \kappa^*)^2\int_{-2}^0 dy\,\mu_{\text{s.c.}}(y)\,y^2\right] \\ &\approx \delta\left[2 - (1 - \kappa^*)^2\right]\,, \end{aligned} \tag{76}$$

implying that the equation for $\delta$ is satisfied for $\delta = 0$, and that
$$\alpha_{\text{inter}} = \frac{1 + 2\kappa^* - \kappa^{*2}}{4}\,. \tag{77}$$

Let us notice that the $\delta = 0$ solution here is valid for all $\alpha > \alpha_{\text{inter}}$, so we could have worked with (6) with $\tilde{\lambda} = 0$ and finite $\epsilon$, and computed $\epsilon$ explicitly as a function of $\alpha$. Doing this shows that $\epsilon < +\infty$ for all $\alpha > \alpha_{\text{inter}}$, and that $\epsilon \to +\infty$ exactly at the threshold. This concludes the derivation of Result 1.

## C.3   Derivation of Corollary 1

Consider first the case $0 < \kappa^* < 1$. We start from the stationary conditions for (6)

$$
\begin{cases}
4\alpha\delta - \frac{\delta}{\epsilon} = \partial_\delta J\left(\delta, \tilde{\lambda}\epsilon\right) \\
Q^* + 2\alpha\delta^2 + \frac{\Delta}{2} - \frac{\delta^2}{\epsilon} = [(1 - s\partial_s)J(\delta, s)]_{s=\epsilon\tilde{\lambda}}
\end{cases}
\tag{78}
$$

To compute the strong recovery, i.e. the value of $\alpha$ at which the test error is zero, we look for the value of $\alpha$ that is consistent with a solution of (6) in the limit $\delta \to 0^+$. We assume the scaling

$$
\epsilon = \delta\sqrt{1 - \kappa^*}c
\tag{79}
$$

and verify it self-consistently. We have that (71) implies at leading order

$$
\partial_\delta \int_{-\infty}^{\tilde{\lambda}\epsilon} dx\, \mu_\delta^*(x)\,(x - \tilde{\lambda}\epsilon)^2
$$

$$
= \partial_\delta \delta^2(1 - \kappa^*) \int_{-\infty}^{\tilde{\lambda}\epsilon/\delta\sqrt{1-\kappa^*}} dy\, \delta\sqrt{1 - \kappa^*}\mu_\delta^*(\delta\sqrt{1 - \kappa^*}y)\left(y - \frac{\tilde{\lambda}\epsilon}{\delta\sqrt{1 - \kappa^*}}\right)^2
$$

$$
\approx \partial_\delta \delta^2(1 - \kappa^*)^2 \int_{-2}^{\tilde{\lambda}\epsilon/\delta\sqrt{1-\kappa^*}} dy\, \mu_{\text{s.c.}}(y)\left(y - \frac{\tilde{\lambda}\epsilon}{\delta\sqrt{1 - \kappa^*}}\right)^2
\tag{80}
$$

$$
= 2\delta(1 - \kappa^*)^2\left(M_{\text{s.c.}}^{(2)}(\tilde{\lambda}c) - 2\tilde{\lambda}cM_{\text{s.c.}}^{(1)}(\tilde{\lambda}c) + \tilde{\lambda}^2c^2 M_{\text{s.c.}}^{(0)}(\tilde{\lambda}c)\right)
$$

$$
+ 2\delta(1 - \kappa^*)^2\left(\tilde{\lambda}cM_{\text{s.c.}}^{(1)}(\tilde{\lambda}c) - \tilde{\lambda}^2c^2 M_{\text{s.c.}}^{(0)}(\tilde{\lambda}c)\right)
$$

$$
= 2\delta(1 - \kappa^*)^2\left(M_{\text{s.c.}}^{(2)}(\tilde{\lambda}c) - \tilde{\lambda}cM_{\text{s.c.}}^{(1)}(\tilde{\lambda}c)\right),
$$

where we defined the incomplete $k$-th moment of the semi-circle distribution as

$$
M_{\text{s.c.}}^{(k)}(x) = \int_{-2}^{x} dy\, \mu_{\text{s.c.}}(y)\, y^k\,.
\tag{81}
$$

Similarly

$$
\left[(1 - s\partial_s)\int_{-\infty}^{s} dx\, \mu_\delta^*(x)\,(x - s)^2\right]_{s=\epsilon\tilde{\lambda}} = \int_{-\infty}^{\tilde{\lambda}\epsilon} dx\, \mu_\delta^*(x)\,(x - \tilde{\lambda}\epsilon)^2
$$

$$
+ 2\epsilon\tilde{\lambda}\int_{-\infty}^{\tilde{\lambda}\epsilon} dx\, \mu_\delta^*(x)\,(x - \epsilon\tilde{\lambda})
\tag{82}
$$

$$
= \int_{-\infty}^{\tilde{\lambda}\epsilon} dx\, \mu_\delta^*(x)\,(x^2 - \tilde{\lambda}\epsilon^2)
$$

$$
\approx \delta^2(1 - \kappa^*)^2\left(M_{\text{s.c.}}^{(2)}(\tilde{\lambda}c) - \tilde{\lambda}^2c^2 M_{\text{s.c.}}^{(0)}(\tilde{\lambda}c)\right).
$$

This leads to the equations (at leading order in $\delta \to 0^+$)

$$
\begin{cases}
\alpha = \frac{1}{2} + \frac{\tilde{\lambda}}{4c\sqrt{1-\kappa^*}\delta} - \frac{1}{2}(1 - \kappa^*)^2\left(M_{\text{s.c.}}^{(2)}(c) - cM_{\text{s.c.}}^{(1)}(c)\right) \\
\alpha = \frac{1}{2} - \frac{\Delta}{4\delta^2} + \frac{\tilde{\lambda}}{2c\sqrt{1-\kappa^*}\delta} - \frac{1}{2}c^2(1 - \kappa^*) - \frac{1}{2}(1 - \kappa^*)^2\left(M_{\text{s.c.}}^{(2)}(c) - c^2 M_{\text{s.c.}}^{(0)}(c)\right)
\end{cases}
\tag{83}
$$

where we renamed $c\tilde{\lambda} \to c$ (which will allow to be in the interpolation limit for $\tilde{\lambda} = 0$ as detailed in Appendix B). Define

$$
\bar{\lambda} = \frac{\tilde{\lambda}}{4\delta\sqrt{1 - \kappa^*}} \quad \text{and} \quad \bar{\Delta} = \frac{\Delta}{4\delta^2},
\tag{84}
$$

that is the small regularization limit and noiseless limit at the critical scaling. Then the equations read

$$\begin{cases} \frac{\bar{\lambda}}{c} - \bar{\Delta} = \frac{1}{2}(1 - \kappa^*)^2 \left( c M_{\text{s.c.}}^{(1)}(c) - c^2 M_{\text{s.c.}}^{(0)}(c) + \frac{c^2}{1-\kappa^*} \right) \\ \alpha_{\text{strong}} = \frac{1}{2} + \frac{\bar{\lambda}}{c} - \frac{1}{2}(1 - \kappa^*)^2 \left( M_{\text{s.c.}}^{(2)}(c) - c M_{\text{s.c.}}^{(1)}(c) \right) \end{cases}, \qquad (85)$$

giving one equation for $c$, and one for $\alpha_{\text{strong}}$ given $c$. By computing the next-to-leading order, one could access the speed at which $\alpha \to \alpha_{\text{strong}}$ in $\delta$, invert it and find the critical scalings of $\tilde{\lambda}$ and $\Delta$ as a function of $\alpha_{\text{strong}} - \alpha$. In the case $\tilde{\lambda} = \Delta = 0$ we have the simpler equation

$$\begin{cases} c = (1 - \kappa^*) \left( c M_{\text{s.c.}}^{(0)}(c) - M_{\text{s.c.}}^{(1)}(c) \right) \\ \alpha_{\text{strong}} = \frac{1}{2} - \frac{1}{2}(1 - \kappa^*)^2 \left( M_{\text{s.c.}}^{(2)}(c) - c M_{\text{s.c.}}^{(1)}(c) \right) \end{cases}, \qquad (86)$$

which modulo conversions gives the same $\alpha_{\text{strong}}$ as [Donoho et al., 2013, 12].

For $\kappa^* > 1$, the same equations hold with all $M$ contributions set to zero (see discussion above), giving (again for $\tilde{\lambda} = \Delta = 0$)

$$\alpha_{\text{strong}} = \frac{1}{2}. \qquad (87)$$

This shows Corollary 1.

## D   Derivation of Result 2

We start again from (6)

$$\begin{cases} 4\alpha\delta - \frac{\delta}{\epsilon} = \partial_\delta J \left( \delta, \tilde{\lambda}\epsilon \right) \\ Q^* + 2\alpha\delta^2 + \frac{\Delta}{2} - \frac{\delta^2}{\epsilon} = J(\delta, \tilde{\lambda}\epsilon) - \epsilon\tilde{\lambda} J^{(0,1)}(\delta, \epsilon\tilde{\lambda}). \end{cases} \qquad (88)$$

We redefine $\tilde{\lambda}\epsilon \to \epsilon$ and consider the limit $\kappa^* \to 0$ under the scaling ansatz $\alpha = \bar{\alpha}\kappa^*$, $\delta = d\kappa^{*-1/2}$, $\epsilon = c\kappa^{*-1/2}$ and $\tilde{\lambda} = \bar{\lambda}\kappa^{*1/2}$. Notice that $Q^* = 1 + \kappa^* \approx 1$ in the limit. The equations read

$$\begin{cases} 4\bar{\alpha}d - \bar{\lambda}\frac{d}{c} = \partial_d J \left( \delta, \epsilon \right) \\ 1 + 2\bar{\alpha}d^2 + \frac{\Delta}{2} - \bar{\lambda}\frac{d^2}{c} = (1 - c\partial_c) J(d\kappa^{*-1/2}, c\kappa^{*-1/2}) \end{cases} \qquad (89)$$

Then

$$\begin{aligned} J \left( d\kappa^{*-1/2}, c\kappa^{*-1/2} \right) &= \int_{c\kappa^{*-1/2}}^{+\infty} dx \, \mu^*_{d\kappa^{*-1/2}}(x) \, (x - c\kappa^{*-1/2})^2 \\ &= \kappa^{*-1} \int_c^{+\infty} dy \, \kappa^{*-1/2} \mu^*_{d\kappa^{*-1/2}}(y\kappa^{*-1/2}) \, (y - c)^2. \end{aligned} \qquad (90)$$

We now use the fact that at leading order for small $\kappa^*$

$$\nu(y) \approx \kappa^{*-1/2} \mu^*_{d\kappa^{*-1/2}}(y\kappa^{*-1/2}) \qquad (91)$$

is given by a semi-circle distribution of radius $2d$ centered at zero with mass $1 - \kappa^*$, plus a Dirac's delta spike at $y = 1 + d^2$ with mass $\kappa^*$ if $d^2 \leq 1$, see [Maillard et al., 2024, Appendix E.1].

We then have the following cases in which the spike does not contribute.

- No spike $d^2 > 1$ and $c > 2d$, or spike $0 < d^2 < 1$ and $c > 1 + d^2 > 2d$. Then the integral $J$ does not contribute to the equations. Thus

$$\begin{cases} 4\bar{\alpha}d = \frac{d}{c}\bar{\lambda} \\ 1 + 2\bar{\alpha}d^2 + \frac{\Delta}{2} - \frac{d^2}{c}\bar{\lambda} = 0 \end{cases} \implies \begin{cases} c = \frac{\bar{\lambda}}{4\bar{\alpha}} \\ d^2 = \frac{1+\Delta/2}{2\bar{\alpha}} \end{cases}, \qquad (92)$$

  under the condition

$$\begin{cases} d^2 > 1 \iff \bar{\alpha} < \frac{1+\Delta/2}{2} \\ c > 2d \iff \bar{\alpha} < \frac{\bar{\lambda}^2}{32(1+\Delta/2)} \end{cases} \implies \bar{\alpha} < \min \left( \frac{1+\Delta/2}{2}, \frac{\bar{\lambda}^2}{32(1+\Delta/2)} \right), \quad (93)$$

  or

$$\begin{cases} d^2 \leq 1 \iff \bar{\alpha} \geq \frac{1+\Delta/2}{2} \\ c > 1 + d^2 \iff \bar{\alpha} < \frac{\bar{\lambda}-2(1+\Delta/2)}{4} \end{cases} \implies \frac{1+\Delta/2}{2} \leq \bar{\alpha} < \frac{\bar{\lambda}-2(1+\Delta/2)}{4}. \quad (94)$$

- No spike $d^2 > 1$ and bulk integral contribution $c \to (2d)^-$. In this case $c$ tends to $2d$ with the appropriate rate in order to have order $\mathcal{O}(1)$ contribution from the bulk integral in the equations. We have, calling $2d - c = dt$ i.e. $c = d(2 - t)$ for some $t > 0$

$$\kappa^{*-1} \int_{(2-t)d}^{2d} dy \, \frac{1}{d} \mu_{\text{s.c.}} \left(\frac{y}{d}\right) (y - d(2 - t))^2 = d^2 \kappa^{*-1} \int_{2-t}^{2} dy \, \mu_{\text{s.c.}}(y) (y - 2 + t)^2$$
$$= d^2 \kappa^{*-1} \frac{16 t^{7/2}}{105\pi} + O\left(\kappa^{*-1} t^{9/2}\right) \tag{95}$$

giving

$$J^{\text{bulk}} = \frac{16 d^2 t^{7/2}}{105\pi\kappa^*} \approx 0$$
$$\partial_d J^{\text{bulk}} = \frac{16 d t^{5/2}}{15\pi\kappa^*} \approx T \tag{96}$$
$$\partial_c J^{\text{bulk}} = -\frac{8 d t^{5/2}}{15\pi\kappa^*} \approx -\frac{T}{2}$$

from which we see that the bulk integral contributes to non-diverging order if $t^{5/2} = \mathcal{O}(\kappa^*)$, giving the equations

$$\begin{cases} 4\bar{\alpha}d - \bar{\lambda}\frac{1}{2} = T \\ 1 + 2\bar{\alpha}d^2 + \frac{\Delta}{4} - \bar{\lambda}\frac{d}{2} = dT \end{cases} \implies \begin{cases} T = 2\sqrt{2\bar{\alpha}(1 + \Delta/2)} - \frac{1}{2}\bar{\lambda} \\ d^2 = \frac{1+\Delta/2}{2\bar{\alpha}} \end{cases} \tag{97}$$

under the conditions

$$\begin{cases} d^2 > 1 \implies \bar{\alpha} < \frac{1+\Delta/2}{2\bar{\alpha}} \\ T > 0 \implies \bar{\alpha} > \frac{\bar{\lambda}^2}{32(1+\Delta/2)} \end{cases} \implies \frac{\bar{\lambda}^2}{32(1 + \Delta/2)} < \bar{\alpha} < \frac{1 + \Delta/2}{2\bar{\alpha}} \tag{98}$$

Thus, we start finding that

$$\text{MSE} = 1 \quad \text{for} \quad \bar{\alpha} < \max\left(\frac{1 + \Delta/2}{2}, \frac{\bar{\lambda} - 2(1 + \Delta/4)}{4}\right). \tag{99}$$

We now consider the case in which the spike contributes.

- If there is the spike $d^2 < 1$ and it is the only contribution $2d < c < 1 + d^2$, then the equations read

$$\begin{cases} 4\bar{\alpha}d - \frac{d}{c}\bar{\lambda} = 4d(1 + d^2 - c) \\ 1 + 2\bar{\alpha}d^2 - \frac{d^2}{c}\bar{\lambda} + \frac{\Delta}{2} = \left(1 + d^2 - c\right)^2 + 2c\left(1 + d^2 - c\right) \end{cases} \tag{100}$$

This system can be solved explicitly under the conditions above, giving quite an unmanageable expression. It reduces to $\delta = 0$ for $\Delta = \bar{\lambda} = 0$.

- If there is the spike $d^2 < 1$ and it contributes along with the bulk (see discussion above and (96)) $c \to (2d)^-$, then the equations read

$$\begin{cases} 4\bar{\alpha}d - \frac{1}{2}\bar{\lambda} = 4d(1 - d)^2 + T \\ 1 + 2\bar{\alpha}d^2 - \frac{d}{2}\bar{\lambda} + \frac{\Delta}{4} = (1 - d)^4 + 4d(1 - d)^2 + dT \end{cases} \tag{101}$$

These give again unmanageable expressions, but for $\Delta = 0$ it gives

$$\text{MSE} = \frac{2}{9}\bar{\alpha}\left(4 - \sqrt{6\bar{\alpha} - 2}\right)^2. \tag{102}$$

One can check by explicit solution that the last two cases lead to test error strictly smaller than one. The last two cases can be easily solve numerically, allowing to plot the curves in Figure 4.

Finally, the large $\alpha$ behavior can be found by considering the last case

$$\begin{cases} 4\bar{\alpha}d - \frac{1}{2}\bar{\lambda} = 4d(1 - d)^2 + T \\ 1 + 2\bar{\alpha}d^2 - \frac{d}{2}\bar{\lambda} + \frac{\Delta}{4} = (1 - d)^4 + 4d(1 - d)^2 + dT \end{cases} \tag{103}$$

under the scaling ansatz

$$d = \frac{1}{\sqrt{\alpha}} \left( d_0 + \frac{d_1}{\alpha} + \dots \right) \quad T = \sqrt{\alpha} \left( T_0 + \frac{T_1}{\sqrt{\alpha}} + \dots \right) \tag{104}$$

and solving the equations perturbatively gives

$$d_0 = \frac{\sqrt{\Delta}}{2} \quad \text{and} \quad d_1 = \frac{3\sqrt{\Delta}}{4} \tag{105}$$

giving the large $\alpha$ scaling presented in Result 2.

This completes the derivation of Result 2.

## E   Details of the numerical implementation of Theorem 1

We provide the code for the numerical implementation of the equations in Theorem 1 and the experiments at https://github.com/SPOC-group/OverparametrisedNet.

The numerical implementation of (6) reduces, at its core, to two sub-tasks: the numerical evaluation of $\mu_a^*$, the integration involved in computing $J(a,b)$ and its partial derivatives, and the numerical solution of the equations themselves.

**Computing $\mu_a^*$.**   Recall that the spectral density $\mu_a^*$ is the free convolution of a rescaled Marchenko-Pastur $\mu^*$ distribution, and that of a semicircle distribution with radius $2a$. In particular, $\mu^*(x) = \sqrt{\kappa^*}\mu_{\text{M.P.}}(\sqrt{\kappa^*}x)$, where $\mu_{\text{M.P.}}$ is the Marchenko-Pastur distribution with parameter $\kappa^*$ (the asymptotic spectral distribution of $A^T A/m$ where $A \in \mathbb{R}^{m \times d}$ has i.i.d. zero-mean unit-variance components). This free convolution can be computed using standard random matrix theory [Potters and Bouchaud, 2020] tools. We report here the procedure as described in Maillard et al. [2024], and summarize here only the steps without providing a derivation.

To compute the spectral density under consideration, first we fix the noise level $a$ and the location $x$ at which we want to compute $\mu_a^*(x)$. Then, we solve the following cubic equation (looking for the solution with largest imaginary part, if all solutions are real then the spectral density at that location is zero)

$$\left( \frac{1}{\sqrt{\kappa^*}}a \right) g^3 - \left( \frac{z}{\sqrt{\kappa^*}} + a \right) g^2 + \left( z + \frac{1}{\sqrt{\kappa^*}} - \sqrt{\kappa^*} \right) g - 1 = 0 \,, \tag{106}$$

where $z = x - i\tau$ with $\tau$ a small positive constant.

The solution $g = g(z)$ of the equation is the so-called Stieltjes transform of $\mu^*(x)$, i.e.

$$g(z) = \int dx \, \frac{\mu(x)}{x - z} \,, \tag{107}$$

which can be inverted by the Stieltjes–Perron inversion formula, i.e.

$$\mu_a^*(x) = \lim_{\tau \to 0^+} \frac{1}{\pi} \operatorname{Im} g_{\mu_Y}(x - i\tau) \,. \tag{108}$$

In practice, the limit is done by using $\tau = 10^{-12}$ directly in (106).

Additionally, by imposing that the discriminant of (106) is zero we obtain an equation for $x$ (setting $\tau = 0$) that allows to precisely evaluate the boundaries of each of the bulks of $\mu_a^*$ (the distribution has two bulks in the low-noise setting, and one single bulk otherwise, with the splitting threshold depending non-trivially on $\kappa^*$ and $a$).

**Computing $J(a,b)$.**   We use the estimate of the bulk boundaries to set-up an efficient integration scheme to finally compute $J(a,b)$. We first compute the bulk boundaries, then intersect the support of the bulks with the integration region $(b, +\infty)$, and finally for each bulk integrate $\mu_a^*(x)(x-b)^2$ using `quad` from the Scipy library in Python. This explicit use of the bulk boundaries allows precise evaluation of $J(a,b)$ even in the low-noise regime, where one of the bulks has very small support with quite large values for the spectral density.

Finally, we compute the partial derivatives of $J(a,b)$ by centered finite differences, with step size equal to $10^{-8}$.

**Numerical solution of the equations.** To solve numerically (6), we find it convenient to revert to the state evolution equations presented in Appendix A.4, equation (59). We follow the prescribed iteration, possibly damping the updates with a factor as small as $10^{-2}$, and declaring convergence when the Euclidean distance between two subsequent updates falls below a prescribed tolerance of at least $10^{-4}$. It is also convenient to adopt a planting scheme, where the equations for a given value of $\tilde{\lambda}, \Delta, \kappa^*$ are solved for one initial value of $\alpha$ (we found that a value around the mid-point between $\alpha = 0$ and $\alpha = \alpha_{\text{strong}}$ works best), and then the rest of the learning curve as a function of $\alpha$ is computed by successively using the solution at a value $\alpha$ as the initialization for that of $\alpha + \delta\alpha$, where $\delta\alpha$ is a small step-size.

## F  Details of the experiments

For Figure 1 right and $d \leq 100$ we used the CVXPY package in Python. All the other experiments are realized in PyTorch. We instantiate the student weights randomly as centered Gaussian variables with standard deviation $10^{-3}$ and the target ones with variance 1, appropriately changing the functional form of the target. In Figure 1 right we optimize using LBFGS for the sake of efficiency. For Figure 1 left we optimize using the GD iteration

$$\mathbf{w}_k^{t+1} = \mathbf{w}_k^t - 2\lambda\eta\frac{\mathbf{w}_k^t}{\sqrt{md}} - \eta\sum_{\mu=1}^{n}\nabla_{\mathbf{w}_k}\left[y_\mu - \frac{1}{\sqrt{m}}\sum_{k=1}^{m}\sigma_k\left(\frac{\mathbf{w}_k^t \cdot \mathbf{x}_\mu}{\sqrt{d}}\right)\right]^2 \tag{109}$$

with $\eta = 20$.

## G  Arbitrary sign in the second layer weights

A key limitation of the network in (1) is that it can only learn targets with positive definite $S^*$. This is because the second layer weights are all positive. Within our formalism we can consider a more generic function with fixed $a_k \in \{-1, 1\}$ with at least $d$ positive and $d$ negative entries.

$$\hat{f}(\mathbf{x}; W) = \frac{1}{\sqrt{m}}\sum_{k=1}^{m}a_k\sigma_k\left(\frac{\mathbf{w}_k \cdot \mathbf{x}}{\sqrt{d}}\right) = \text{Tr}\left(X(\mathbf{x})\frac{1}{\sqrt{dm}}\sum_{i=1}^{m}a_i\mathbf{w}_i\mathbf{w}_i^T\right), \tag{110}$$

with loss

$$\tilde{\mathcal{L}}_W(W) := \sum_{\mu=1}^{n}\text{Tr}\left(X(\mathbf{x})\left(S^* - \frac{1}{\sqrt{dm}}\sum_{i=1}^{m}a_i\mathbf{w}_i\mathbf{w}_i^T\right)\right)^2 + \lambda\sum_{i=1}^{m}\|\mathbf{w}_i\|^2 \tag{111}$$

where

$$S^* = \frac{1}{\sqrt{dm^*}}\sum_{i=1}^{m^*}a_i^*\mathbf{w}_i^*(\mathbf{w}_i^*)^T. \tag{112}$$

Our goal will be to show that minimizing over $\tilde{\mathcal{L}}_W(W)$ is equivalent to minimize $\tilde{\mathcal{L}}_S(S)$ from (9) over all symmetric matrices $S$

$$\hat{S} = \arg\min_{S \in \text{Sym}(d)} \tilde{\mathcal{L}}_S(S), \text{ with } \tilde{\mathcal{L}}_S(S) := \sum_{\mu=1}^{n}\text{Tr}\left[\frac{\mathbf{x}_\mu\mathbf{x}_\mu^T - \mathbb{I}_d}{\sqrt{d}}(S - S^*)\right]^2 + \sqrt{md}\lambda\|S\|_*. \tag{113}$$

The argument goes as follows. For any fixed value of $a$, define the sets of indices $P, N$ as

$$P = \{k|a_k > 0\}, \qquad N = \{k|a_k < 0\}, \tag{114}$$

and additionally, define

$$(S_P)_{ij} = \frac{1}{\sqrt{md}}\sum_{k\in P}\hat{W}_{ki}\hat{W}_{kj}, \qquad (S_N)_{ij} = -\frac{1}{\sqrt{md}}\sum_{k\in N}\hat{W}_{ki}\hat{W}_{kj}. \tag{115}$$

The matrix $S$ is then given by the sum $S = S_P + S_N$, and we remark that both matrices are symmetric, non-rank constrained and respectively positive and negative semi-definite. The minimization of $\tilde{\mathcal{L}}_W(W)$ over $W$ can be rewritten as

$$(\hat{S}_P, \hat{S}_N) = \underset{\substack{S_P \succeq 0 \\ S_N \preceq 0}}{\arg\min}\, \tilde{\mathcal{L}}_{PN}(S_P, S_N),$$

$$\text{with } \tilde{\mathcal{L}}_{PN}(S_P, S_N) := \sum_{\mu=1}^{n} \mathrm{Tr}\left[Z^\mu (S_P + S_N - S^*)\right]^2 + \sqrt{md}\lambda(\|S_P\|_* + \|S_N\|).$$

(116)

What we have to show is that (113) and (116) have the same optimal value, and that the minimizer $\hat{S}$ of (113) yields a minimizer $(\hat{S}_P, \hat{S}_N)$ of (116) by taking the positive and negative spectral parts of $\hat{S}$. First, we write

$$\min_{\substack{S_P \succeq 0 \\ S_N \preceq 0}} \tilde{\mathcal{L}}_{PN}(S_P, S_N) = \min_{S \in \mathrm{Sym}(d)} \sum_{\mu=1}^{n} \mathrm{Tr}\left[\frac{\mathbf{x}_\mu \mathbf{x}_\mu^T - \mathbb{I}_d}{\sqrt{d}} (S - S^*)\right]^2 + \sqrt{md}\lambda R(S) \qquad (117)$$

where

$$R(S) := \inf_{\substack{S_P \succeq 0,\, S_N \preceq 0 \\ S_P + S_N = \bar{S}}} \left(\|S_P\|_* + \|S_N\|_*\right). \qquad (118)$$

By the triangle inequality

$$\|S\|_* = \|S_P + S_N\|_* \le \|S_P\|_* + \|S_N\|_*, \qquad (119)$$

hence $R(S) \ge \|S\|_*$. This bound is in fact achievable: let $S$ be symmetric, with spectral decomposition

$$S = U\,\mathrm{diag}(\sigma_1, \ldots, \sigma_d)\,U^\top. \qquad (120)$$

Define its positive and negative spectral parts

$$S_+ := U\,\mathrm{diag}(\sigma_i^+)\,U^\top, \qquad S_- := U\,\mathrm{diag}(\sigma_i^-)\,U^\top, \qquad (121)$$

where

$$\sigma_i^+ := \max(\sigma_i, 0), \qquad \sigma_i^- := \min(\sigma_i, 0) \le 0. \qquad (122)$$

Then $S_+ \succeq 0$, $S_- \preceq 0$, and $S = S_+ + S_-$. Moreover,

$$\|S_+\|_* = \sum_{i:\, \sigma_i > 0} \sigma_i^+, \qquad \|S_-\|_* = \sum_{i:\, \sigma_i < 0} (-\sigma_i^-), \qquad (123)$$

so

$$\|S_+\|_* + \|S_-\|_* = \sum_i |\sigma_i| = \|S\|_*.$$

Thus, for this admissible decomposition $R(S) \le \|S\|_*$, which combined with the other inequality gives us $R(S) = \|S\|_*$. This proves the claim.

**Asymptotics without PSD constraint.** We are solving the same problem as (22), just without the positivity constraint:

$$\hat{S} = \underset{S \in \mathrm{Sym}(d)}{\arg\min}\, \tilde{\mathcal{L}}(S), \text{ with } \tilde{\mathcal{L}}(S) := \sum_{\mu=1}^{n} \mathrm{Tr}\left[\frac{\mathbf{x}_\mu \mathbf{x}_\mu^T - \mathbb{I}_d}{\sqrt{d}} (S - S^*)\right]^2 + \sqrt{md}\left(\lambda\|Tr(S\|_*) + \tau\|S\|_F^2\right).$$

(124)

For this case, the spectral denoising function (48) becomes

$$L_i = \frac{1}{2\tau + k}\mathrm{SoftThresholding}\left(D_i, 2\tilde{\lambda}\right), \qquad (125)$$

where

$$\mathrm{SoftThresholding}(x, \theta) = \mathrm{sign}(x)\max\{|x| - \theta, 0\}. \qquad (126)$$

Similarly, equation (57) needs to be substituted by

$$L_i = \frac{m_u}{2\tau + k}\mathrm{SoftThresholding}\left(D_i, 2\tilde{\lambda}/m_u\right). \qquad (127)$$

The final state evolution will be unchanged by redefining $J$ in (6) as

$$J(a, b) = \int_{|x| > b} dx\, \mu_a^*(x)\,(|x| - b)^2. \qquad (128)$$

We illustrate how this changes the theoretical predictions in Figure 5.

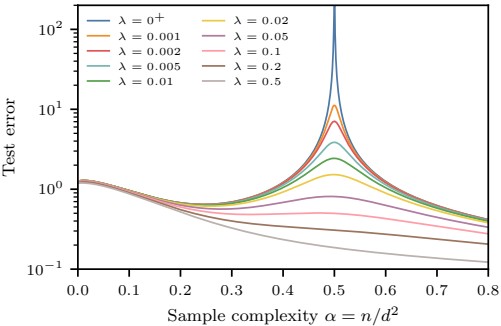

Figure 5: Test error as a function of the sample complexity $\alpha$ for different values of the regularization when optimizing without PSD constraint as described in G. Here $\kappa^* = 0.2$ and $\Delta = 0.5$. Notice that in this case the interpolation peak is at $\alpha = 0.5$, and there the test error diverges.

