# OpenReview forum: "The Nuclear Route: Sharp Asymptotics of ERM in Overparameterized Quadratic Networks"
_NeurIPS.cc/2025/Conference — NeurIPS 2025 poster_

### Official Review · Reviewer_5q53 · 2025-06-08

**Clarity:** 2
**Significance:** 3
**Originality:** 3
**Rating:** 4
**Confidence:** 3

**Summary:**

The authors analyze a two-layer, overparameterized neural network with quadratic activation, trained by empirical risk minimization (ERM) on data generated by the same model. They show that, in the high-dimensional limit (input dimension $d$, number of samples $n$, network width $m$ all  $\rightarrow \infty$ at fixed ratios), the nonconvex ERM problem can be exactly reformulated as a convex nuclear-norm‐penalized positive semidefinite matrix sensing problem. Leveraging Gaussian universality, AMP/state‐evolution, and convex‐optimization tools, they derive closed-form asymptotic expressions for the training error, test error, and singular‐value distribution of the learned weight matrix. Their formulas reveal sharp thresholds for perfect recovery, interpolation, and the double‐descent phenomenon as functions of sample complexity $d^2$, width ratio $\kappa$, regularization strength $\lambda$, and noise level $\Delta$.

**Questions:**

1. I want to make sure if the training and testing errors in (7) can be always non-negative?

2. Standard GAMP / AMP state‐evolution theory covers separable scalar denoisers. Here your denoiser acts on symmetric matrix blocks. What modifications to the state‐evolution proof are needed to handle this non‐separable, PSD‐projected “denoiser”? Are there additional convergence or stability conditions?

3. In the unregularized case ($\lambda=0$) you identify an interpolation threshold αinterαinter​. Is it possible that local but non-global minima exist just below $\alpha_{inter}$?

4. For the low rank setting $\kappa^\star <<1$, can you compare learning performance with work [1]?
[1] Li et al. (COLT 2018) in “Algorithmic Regularization in Over-parameterized Matrix Sensing and Neural Networks with Quadratic Activations.”

**Ethical Concerns:**

["NO or VERY MINOR ethics concerns only"]

**Final Justification:**

I keep my recommendation at 4 (Borderline accept). The paper provides sharp, explicit asymptotics for ERM in quadratic two-layer networks via a clean convex reformulation and AMP/SE analysis. After the rebuttal and discussion, several of my concerns were addressed; a few remain open but do not outweigh the technical contribution.

**Limitations:**

1. Authors study the well-specified student-teacher model and quadratic activation,  the more realistic setting is considering mis-specified case where student and teacher have different models and general activation functions beyond quadratic and smooth activations such as widely used ReLU-type activation functions.

2. All sharp predictions are exact as $d,n,m \rightarrow \infty$. No finite-$d$ error bounds or convergence rates are provided. For moderate dimensions, it’s unclear how large the bias or variance of the asymptotic formulas is.

3. Limited insight on feature learning. By focusing on a fixed quadratic feature map, the paper sidesteps any notion of learning useful representations from raw data. All the heavy lifting happens in the convex PSD stage, with no mechanism for adapting or discovering features—a key strength of deep networks in practice.

**Quality:**

3

**Strengths And Weaknesses:**

**Strengths**
1. Exact asymptotics: Delivers fully explicit high-dimensional limits for both training and generalization error—stronger than typical “order-level” bounds.

2. Convex reformulation: The reduction from a nonconvex quadratic network to a convex nuclear-norm problem is conceptually elegant and may inspire similar mappings for other architectures.

3. Universality: By invoking one-dimensional CLT and an interpolation argument, they show their results hold for a broad class of input distributions, not just Gaussian.

4. Numerical Validation: Experiments in moderate dimensions (e.g. $d=50,100,300$) match the theory closely, demonstrating practical relevance of the asymptotics.

**Weaknesses**
1. Model specificity: Focuses exclusively on centered quadratic activations; it is unclear how to adapt to ReLU, tanh, or deeper nets.

2. Synthetic data: All experiments use rotationally invariant Gaussian-like data; performance on real-world datasets remains untested.

3. Regularization dependence: The sharp formulas require nonzero nuclear regularization $\lambda>0$; the purely unregularized case is only partially characterized by an interpolation threshold. Actually, many existing works focus on learning theory of regularization-free models that demonstrate the optimization methods can automatically found the parameters with good generalization performance from multiple solutions.

4. Not well-defined notations. For example, authors used semicircle distribution and free convolution terms in Theorem 1 without introduced beforehand. The $\delta$ in defined  radius of semicircle distribution conflicts with the $\delta$ function in Eq. (8).

---

> ### Author Rebuttal · Authors · 2025-07-30
>
> We thank the reviewer for their constructive comments.
>
> **W1** We agree and acknowledge this in the conclusion. Generic activation is an open challenge. See also the answer to referee 8thr.
>
> **W2**  The contribution of our paper is a theoretical insight into the mechanisms of feature learning under overparametrization. Single-hidden-layer quadratic networks have limited expressive power and thus their performance on real datasets will not be very good. It would, however, be interesting to construct models for the data structure that would reproduce the learning curves on real datasets. We will add a comment on this point.
>
> **W3**
> To study which solution is found in the unregularized case in the interpolation regime, a sharp analysis of the gradient descent algorithm would be needed. This is a challenging open problem even in simpler settings and is beyond the scope of the present paper. The regularisation ensures that the ERM is well defined and leads to interesting and new insights. One can take the limit of small regularisation in our equations (appendix B), but in general it will not give the same result as the completely regularisation-free case.
>
> For large enough sample complexity, when the network cannot interpolate anymore, the small regularisation limit accurately describes the minima of the unregularised loss.
>
> **W4** We will fix the notation issues, and we thank them for spotting them.
>
>
> **Q1** Yes, the quantities in equation (7) represent squared errors (training and test), and are therefore non-negative by construction.
>
> **Q2** Indeed one must uses non-separable denoisers in AMP. Surprisingly, this does not introduce much difference, which is one of the stenghts of the AMP formalism. This can be done following the framework of Berthier et al. [2020], which rigorously extends AMP to non-separable denoisers, including those acting on matrix-valued inputs. The key requirement is that the denoising function is Lipschitz, which holds in our case: the denoiser is a soft-thresholding operator on the singular values (Appendix A.4.3). Furthermore, Appendix A.5 outlines a sufficient condition for point-wise convergence of AMP in this setting, which is always satisfied for our problem.
>
> **Q3** In the overparametrized case, Venturi et al showed the absence of spurious minima.
>
> **Q4** Comparison with Li et al. (COLT 2018): Thank you for highlighting this relevant work—we will cite and briefly discuss it in a revised version. In the low-rank regime, our results yield a sharper threshold: as shown in Figure 3 (right) and Equation (17), recovery is possible with $n \sim d m$ samples. This improves upon the scaling $n \sim d m^2$ established in Li et al. for similar models, highlighting a more favorable sample complexity for the overparameterized setting with nuclear norm bias.
>
> **L1** In our work, we consider the mis-specified case where the student and teacher have different models (the student is wider than the dimension, while the teacher's width is varied). The activation being quadratic is indeed an important limitation, but going beyond it is a widely open problem, see also answer to referee  8thr.
>
> **L2** Finite-size accuracy of asymptotic predictions: As shown in Figure 1, the asymptotic predictions already match simulations remarkably well even at very moderate dimensions (e.g., $d = 50$). While we do not provide formal finite-size error bounds, the empirical agreement suggests that bias and variance are small, and likely decay rapidly with dimension. Deriving such bounds rigorously (e.g., along the lines of Rush and Venkataramanan, 2018) is an interesting direction for future work. We do, however, provide rates in the low rank regime in eq.(19) (i.e. the generalization decays as $n^{-1}$.
>
> **L3**: We respectfully but strongly disagree with the reviewer’s assessment. A key contribution of our work is to show that feature learning does occur, and we characterize it precisely. Specifically, we demonstrate that ERM training in this regime induces a nuclear norm bias on the learned feature matrix, as opposed to the Frobenius norm regularization that arises in lazy or NTK-like settings.
>
> This distinction is crucial: just as in matrix compressed sensing, Frobenius regularization ignores low-rank structure and fails to exploit sparsity, while nuclear norm regularization leads to Bayes-optimal recovery. In our setting, this is reflected in Result 2 and Equation (19), where the nuclear-norm–biased ERM achieves optimal test error rates, while NTK-type lazy training would yield near-random performance.
>
> We will add a comment in the revised version to highlight this important distinction and clarify how our work explicitly connects feature learning with training beyond the lazy regime.

---

> > ### Comment · Reviewer_5q53 · 2025-08-02
> >
> > Thank you for your detailed response and pointing out my misunderstanding for the feature learning aspect of your work. I recommend adding the explanation above to the revision to improve clarity

---

> > > ### Author Response · Authors · 2025-08-02
> > >
> > > Thank you very much for pointing out how implicit we were about this key point. We will certainly stress it in the revised version.
> > >
> > > Thank you for helping us improve the clarity of our paper.

---

### Official Review · Reviewer_vWPi · 2025-06-28

**Clarity:** 2
**Significance:** 3
**Originality:** 3
**Rating:** 5
**Confidence:** 3

**Summary:**

The paper analyzes the training and test error of regularized ERM solutions of the empirical square loss for two-layer networks with quadratic activations. They assume that the teacher network is also a quadratic two-layer network, and allow for label noise. They focus on the high-dimensional limit with quadratically many samples and certain bounds on the widths. They provide closed-form expressions for the training and test losses. They use their result to analyze some more specific settings and properties of the solution.

**Questions:**

See the question in the strengths and weaknesses section.

**Ethical Concerns:**

["NO or VERY MINOR ethics concerns only"]

**Final Justification:**

I maintain my original evaluation.

**Limitations:**

Yes

**Quality:**

3

**Strengths And Weaknesses:**

The paper analyzes a basic setting and proves strong and highly non-trivial results for the high-dimensional limit. I believe the results are significant and contribute to our understanding of the problem.

Could the authors also compare their results with the following paper? “On the Power of Over-parametrization in Neural Networks with Quadratic Activation”, Du and Li (2018)

Overall, other than the above question and the obvious limitations mentioned in the last paragraph of the paper, I don’t have further comments. Indeed, the paper considers a rather specific setting with many assumptions, but I still believe that a detailed analysis of this setting is of interest.

---

> ### Author Rebuttal · Authors · 2025-07-30
>
> We thank the reviewer for the positive assessment and for pointing out the relevant work by Du and Li (2018). While our focus is quite different—centered on a precise asymptotic characterization of the global minima of ERM in the high-dimensional regime—we agree that their analysis of the optimization landscape of overparameterized quadratic networks is a valuable and complementary perspective.
>
> Their work studies convergence guarantees of gradient descent in the infinite sample regime and under assumptions that ensure benign landscapes. In contrast, our work focuses on finite-sample generalization, spectral properties, and test error asymptotics in the high-dimensional setting with quadratically many samples, and does not address convergence per se. As such, the goals and techniques are distinct, but we agree this is relevant related work and will include a proper citation and brief discussion in the revised manuscript.

---

> > ### Comment · Reviewer_vWPi · 2025-08-01
> >
> > Thank you for your response

---

### Official Review · Reviewer_aFup · 2025-07-02

**Clarity:** 3
**Significance:** 3
**Originality:** 2
**Rating:** 4
**Confidence:** 3

**Summary:**

This study presents an analysis of empirical risk minimization over the quadratic loss for a two-layer neural network with extensive width and quadratic activations. The authors rigorously derive predictions for the generalization error in high dimensions. The results are then analyzed and used to derive thresholds for interpolation and strong recovery. Finally, all results are placed in the context of related literature on different models and compared with previous Bayes-optimal baselines for the same model.

**Questions:**

### Questions
- What are the key technical differences in the proofs that distinguish the ERM setting considered here from the Bayes-optimal analyses by Maillard et al. [2024] and Xu et al. [2025]?
- Can you comment on the gap between Bayes-optimal performance and the best ERM error in Figure 1 and Figure 2 (left plots)? Does this suggest the presence of a statistical-to-computational gap?
- In line 268, you mention "reflecting the structural advantage of the data over random labeling," but it is unclear to me how the previous reasoning on $\Delta$ and $\kappa^*$ implies this. Could you elaborate further?

### Minor Corrections
- Line 196: "then," not "than."
- In equation (13), the cases for $\kappa^* $  repeat the same condition. I assume the bottom line should state $\kappa^* \ge 1$.

**Ethical Concerns:**

["NO or VERY MINOR ethics concerns only"]

**Final Justification:**

The authors answered clearly to all my quesitons. Stil, I believe that the fact that the setup is very similar to reference Maillard et al. 2024 should be counted, slightly limiting the originality from the modeling poin of view.  Otherwise my rating would be 5, since in all the other aspects I believe this work is very interesting and analyzes thoroughly a particular phenomenology.

**Limitations:**

Yes

**Paper Formatting Concerns:**

No formatting issues.

**Quality:**

4

**Strengths And Weaknesses:**

### Strengths
- The wide network setting is considered to be challenging.
- The setting of Theorem 1 is very general: very few assumptions on the spectral density of $S^*$ are required, providing versatility for different applications.
- This work shows that $\ell^2$ regularization over $W$ translates into nuclear norm regularization over $S$. This correspondence of regularizations is completely new to me.
- The rigorous delineation of thresholds for interpolation and strong recovery is an important contribution to understanding networks with extensive structure. The fact that these thresholds differ from the kernel ones when $\kappa^*$ is small is noteworthy.
- The results in the paper disprove a conjecture on interpolation thresholds from [Sarao Mannelli et al., 2020].

### Weaknesses
- The assumption of i.i.d. Gaussian noise is very restrictive, although it is shared with most of the related literature.
- The quadratic activation assumption is restrictive. However, given the other challenges this model presents, together with the fact that several related works share this assumption, I believe this limitation is not too impactful.
- Several works on this model have already appeared, especially the Bayes-optimal analyses by Maillard et al. [2024] and Xu et al. [2025]. While it is clear that ERM is a different problem from Bayes-optimal analysis, the two settings are related and share some core analytical ideas. In my view, this slightly impacts the originality and significance score.

---

> ### Author Rebuttal · Authors · 2025-07-30
>
> We thank the reviewer for their constructive comments.
>
> **Q1**: While the target function considered is the same as in the Bayes-Optimal setting, the nature of the problem is fundamentally different. Maillard et al. and Xu et al. analyze the Bayes-optimal performance, i.e., the minimum test error achievable by any estimator. Their results do not pertain to neural networks per se, but rather to the theoretical limits of inference. In contrast, our work asks: What does a neural network trained via empirical risk minimization (ERM) actually learn?
>
> From a technical standpoint, although both approaches leverage the Gaussian universality principle, the rest of the analysis is entirely different. Bayes-optimal proofs benefit from powerful interpolation techniques and concentration inequalities that are available in the information-theoretic setting. These tools are not applicable in our case.
>
> To handle the ERM setting, we map the original non-linear optimization to a convex matrix compressed sensing problem and analyze it via Approximate Message Passing (AMP) with non-separable denoisers. This AMP is specifically designed so that its fixed point coincides with the solution of the ERM, thus providing a rigorous and constructive characterization of both generalization error and spectral properties of the learned weights. This approach also allows us to analyze the role of overparameterization, which is inaccessible through purely Bayes-optimal tools.
>
> **Q2**:  This gap does not indicate a statistical-to-computational gap in the usual sense. Rather, it reflects the difference between the information-theoretically optimal estimator (which assumes full knowledge of the model, including the rank or spectrum of the target function) and the ERM solution, which is constrained by the use of a generic $\ell_2$ (or equivalently, nuclear norm) regularization.
>
> In essence, the Bayes-optimal algorithm exploits prior knowledge about the structure of the teacher—most notably, its rank—and can adapt perfectly. The ERM solution instead imposes a generic form of spectral regularization without access to such information. Despite this, we show that ERM achieves nearly optimal performance, often surprisingly close to the Bayes-optimal curve. But some gap is inevitable due to the mismatch in available information and inductive bias.
>
> **Q3**:  This refers to the fact that the interpolation threshold depends non-trivially on the structure of the target function. When $\kappa^* > 0$ and $\Delta < \infty$, the dataset is not unstructured: it contains meaningful correlations inherited from the structure of the teacher. This makes it easier to fit the data than in the worst-case setting of random labels. As a result, interpolation becomes possible at a lower sample complexity than $d^2/2$, when the number of samples equals the numbers of free parameters of the network, which would be the threshold in the random-label scenario.

---

> > ### Comment · Reviewer_aFup · 2025-08-02
> >
> > Thanks for the answers, I confirm my rating.

---

### Official Review · Reviewer_8thr · 2025-07-02

**Clarity:** 1
**Significance:** 2
**Originality:** 3
**Rating:** 3
**Confidence:** 1

**Summary:**

The paper delivers a technically rigorous analysis of two‐layer quadratic networks trained by $\ell_2$‐regularized empirical risk minimization on high‐dimensional Gaussian data, by showing that the nonconvex learning problem can be exactly reformulated as a convex, nuclear‐norm–penalized matrix sensing task. Through precise AMP state‐evolution arguments, the author obtain closed‐form expressions for training and generalization errors in the limit $\alpha = n/d^2 = O(1)$, $m^\star/d = O(1)$ and $m/d = O(1)$, where $n$ is the number of measurements, $d$ is the embedding dimension, $m$ is the number of neurons (i.e., rank of the weight matrix) in the learner's (i.e., student) model and $m^\star$ is the same for the teacher model.

The main results are as follows,

1. The calculation of an interpolation threshold $\alpha^\star_1$ in the case of an i.i.d. prior on $w^\star$ (referred to as the MP case), which corresponds to a characterization of the ratio $\alpha$ for which interpolators collapse to a single solution as the nuclear norm regularization strength goes to $0$. This is a statement about the optimization landscape of the regularized $\ell_2$-regularized empirical risk.

2. The calculation and a strong recovery threshold, $\alpha^\star_2$ in the MP case, beyond which perfect generalization occurs in the noiseless setting. This is shown to be $1/2$ when $m^\star/d \ge 1$ (overparameterized setting).

Their results demonstrate that $\ell_2$ weight decay implicitly enforces low‐rank structure and also results in test error that is independent of $m$ in the overparameterized setting.

**Questions:**

N/A

**Ethical Concerns:**

["NO or VERY MINOR ethics concerns only"]

**Limitations:**

The paper is restricted to the case of quadratic activations and a single layer. While I do not view these as weaknesses of the paper, I think it would be nice if the authors addressed what happens if the activation function is no longer quadratic, for instance.

**Quality:**

2

**Strengths And Weaknesses:**

I am not an expert in this area, and am reviewing this paper as a member of the broader audience. Perhaps someone more well versed with the statistical physics analyses of this problem and mean-field approximations for matrix completion in the overparameterized regime may have a different take about this, but overall I thought that the paper was incredibly hard to read. The main result is quite dense, and missing many simple interpretations. It is written for an expert who already understands the setting (which is admittedly very specific). There are several notations (for instance, free convolution) which were not defined anywhere in the paper, which I believe is necessary for the results to be understood in a self contained manner. Overall, I would recommend that the paper be simplified and presented in a top-down approach, and even potentially lengthened for the goal of making it easier to motivate the main result. I believe my review does not adequately address the technical contribution of the paper, which I was unable to grasp fully. However, improving the writing in the paper seems to be a prerequisite to be able to carry this out.

Minor:

1. $\kappa^\star$ is denoted $\kappa_\star$ in some locations.

---

> ### Author Rebuttal · Authors · 2025-07-30
>
> We thank the reviewer for their thoughtful comments and for engaging with the paper "as a member of the broader audience".
>
> We acknowledge that the paper is technical and that some of the mathematical tools—such as free convolution and matrix compressed sensing—may be unfamiliar to non-specialists. However, the analysis of Hermite expansions, polynomial networks, and student-teacher models with quadratic activations is a well-established and active line of work in theoretical machine learning. Our references include numerous recent contributions in this area.
>
> That said, we appreciate the reviewer’s suggestion and agree that the exposition can be improved in places. In particular, we will clarify the definitions of specialized terms and notations such as free convolution and better motivate our main result in Section 2. We will also fix notational inconsistencies (e.g., the conflict between $\phi$ and $\varphi$, the $\kappa^*$).
>
> As for the question of generalizing beyond quadratic activations: we agree that this is a natural direction, but it's important to highlight that the quadratic case is a necessary prerequisite. The quadratic activation allows for an exact mapping to a convex matrix sensing problem, which underpins our closed-form results. Moving to more general activations—e.g., higher-order monomials—would correspond to a tensor compressed sensing, which is a significantly harder problem for which sharp asymptotic results remain largely open. This is the subject of ongoing work. Nonetheless, the study of quadratic networks is not only tractable but also a classical and insightful approach to understanding nonlinear learning in overparameterized models.
>
> Regarding the broader significance, we would like to emphasize that our analysis sheds light on how feature learning, low-rank bias, and generalization thresholds emerge in overparameterized networks beyond the NTK/lazy regime. While the setting is stylized (quadratic activations, Gaussian inputs), it allows us to rigorously distinguish between regimes where learning occurs and where it does not—something that remains difficult in more general settings. We will add more discussion about these aspects.

---

### Decision · Program_Chairs · 2025-09-17

**Decision:**

Accept (poster)

**Comment:**

This manuscript provides thorough analysis of a simple two layer neural network with quadratic activations trained on synthetic data. In particular, sharp results are provided for both training and test error in specific limits (related to overparametrization). Simple numerical experiments highlight that the theory is illustrative.

The main strength of this manuscript is its sharp characterization of the target model: the theory it provides leads to many interesting follow up observations about the model behavior.

Conversely, the primary weakness of the manuscript is its highly restrictive model and data generation process. In addition, aspects of the presentation could be improved.

Ultimately, the strength of the theoretical results and insights that follow outweigh the restrictiveness of the model. Results of this type are, for the moment, likely limited to more restricted settings. The hope is that the provide stepping stones to further analysis of more realistic models.